Manuscript prepared for Atmos. Chem. Phys.
with version 2015/04/24 7.83 Copernicus papers of the LaTeX class copernicus.cls.
Date: 23 May 2016

# The impact of lightning on tropospheric ozone chemistry using a new global lightning parametrisation

D. L. Finney[1], R. M. Doherty[1], O. Wild[2], and N. L. Abraham[3,4]

[1]School of GeoSciences, The University of Edinburgh, Edinburgh, UK
[2]Lancaster Environment Centre, Lancaster University, Lancaster, UK
[3]Department of Chemistry, University of Cambridge, Cambridge, UK
[4]National Centre for Atmospheric Science, University of Cambridge, Cambridge, UK

*Correspondence to:* D. L. Finney (d.finney@ed.ac.uk)

**Abstract.** A lightning parametrisation based on upward cloud ice flux is implemented in a chemistry-climate model (CCM) for the first time. The UK Chemistry and Aerosols model is used to study the impact of these lightning nitric oxide (NO) emissions on ozone. Comparisons are then made between the new ice flux parametrisation and the commonly-used, cloud-top height parametrisation. The ice flux approach improves the simulation of lightning and the temporal correlations with ozone sonde measurements in the middle and upper troposphere. Peak values of ozone in these regions are attributed to high lightning NO emissions. The ice flux approach reduces the overestimation of tropical lightning apparent in this CCM when using the cloud-top approach. This results in less NO emission in the tropical upper troposphere and more in the extratropics when using the ice flux scheme. In the tropical upper troposphere the reduction in ozone concentration is around 5-10%. Surprisingly, there is only a small reduction in tropospheric ozone burden when using the ice flux approach. The greatest absolute change in ozone burden is found in the lower stratosphere suggesting that much of the ozone produced in the upper troposphere is transported to higher altitudes. Major differences in the frequency distribution of flash rates for the two approaches are found. The cloud-top height scheme has lower maximum flash rates and more mid-range flash rates than the ice flux scheme. The initial $O_x$ (odd oxygen species) production associated with the frequency distribution of continental lightning is analysed to show that higher flash rates are less efficient at producing $O_x$; low flash rates initially produce around 10 times more $O_x$ per flash than high-end flash rates. We find that the newly implemented lightning scheme performs favourably compared to the cloud-top scheme with respect to simulation of lightning and tropospheric ozone. This alternative lightning scheme shows spatial and temporal differences in ozone chemistry which may have implications for comparison between models and observations, and for simulation of future changes in tropospheric ozone.

# 1 Introduction

Lightning is a key source of nitric oxide (NO) in the troposphere. It is estimated to constitute around 10% of the global annual NO source (Schumann and Huntrieser, 2007). However, lightning has particular importance because it is the major source of NO directly in the free troposphere. The oxidation of NO forms $NO_2$ and the sum of these is referred to as $NO_x$. In the middle and upper troposphere $NO_x$ has a longer lifetime and a disproportionately larger impact on tropospheric

chemistry than emissions from the surface.

     Through oxidation, NO is rapidly converted to $NO_2$ until an equilibrium is reached. $NO_2$ photolyses and forms atomic oxygen which reacts with an oxygen molecule to produce ozone, $O_3$. As a source of atomic oxygen, $NO_2$ is often considered together with $O_3$ as odd oxygen, $O_x$. Ozone acts as a greenhouse gas in the atmosphere and is most potent in the upper troposphere where tempera-

ture differences between the atmosphere and ground are greatest (Lacis et al., 1990; Dahlmann et al., 2011). Understanding lightning NO production and ozone formation in this region is important for determining changes in radiative flux resulting from changes in ozone (Liaskos et al., 2015).

     As reported by Lamarque et al. (2013), the parametrisation of lightning in chemistry transport and chemistry-climate models (CCMs) most often uses simulated cloud-top height to determine the flash

rate as presented by Price and Rind (1992). However, this and other existing approaches have been shown to lead to large errors in the distribution of flashes compared to lightning observations (Tost et al., 2007). Several studies have shown that the global magnitude of lightning $NO_x$ emissions is an important contributor to ozone and other trace gases especially in the upper tropical troposphere (Labrador et al., 2005; Wild, 2007; Liaskos et al., 2015). Each of these studies uses a single horizon-

tal distribution of lightning so the impact of varying the lightning emission distribution is unknown. Murray et al. (2012, 2013) have shown that constraining simulated lightning to satellite observations results in a shift of activity from the tropics to extratropics, and that this constraint improves the representation of the ozone tropospheric column and its interannual variability. Finney et al. (2014) showed using reanalysis data that a similar shift in activity away from the tropics occurred when a

more physically based parametrisation based on ice flux was applied.

     The above studies and also that of Grewe et al. (2001) find that the largest impact of lightning emissions of trace gases occurs in the tropical upper troposphere. This is a particularly important region because it is the region of most efficient ozone production (Dahlmann et al., 2011). Understanding how the magnitude of lightning flash rate or concentration of emissions affects ozone production is

an ongoing area of research, and so far has focussed on individual storms or small regions (Allen and Pickering, 2002; DeCaria et al., 2005; Apel et al., 2015). DeCaria et al. (2005) found that whilst there was little ozone enhancement at the time of the storm, there was much more ozone production downstream in the following days. They found a clear positive relationship between downstream ozone production and lightning $NO_x$ concentration which was linear up to $\sim 300$ pptv but resulted

in smaller ozone increases for $NO_x$ increases above this concentration. Increasing ozone produc-

tion downstream with more $NO_x$ was also found by Apel et al. (2015). Allen and Pickering (2002) specifically explored the role of the flash frequency distribution on ozone production using a box model. They found that the cloud-top height scheme produces a high frequency of low flash rates which are unrealistic compared to the observed flash rate distribution. This results in lower $NO_x$

concentrations and greater ozone production efficiency with the cloud-top height scheme. Differences in the frequency distribution between lightning parametrisations were also found across the broader region of the tropics and subtropics by Finney et al. (2014). The importance of differences in flash rate frequency distributions to ozone production over the global domain remains unknown.

In this study, the lightning parametrisation developed by Finney et al. (2014) which uses upward

cloud ice flux at 440 hPa is implemented within the United Kingdom Chemistry and Aerosols model (UKCA). This parametrisation is closely linked to the Non-Inductive Charging Mechanism of thunderstorms (Reynolds et al., 1957) and was shown to perform well against existing parametrisations when applied to reanalysis data (Finney et al., 2014). Here the effect of the cloud-top height and ice flux parametrisations on tropospheric chemistry is quantified using a CCM, focussing especially on

the location and frequency distributions. Section 2 describes the model and observational data used in the study. Section 3 compares the simulated lightning and ozone concentrations to observations. Section 4 analyses the ozone chemistry through use of $O_x$ budgets. Section 5 then considers the differences in zonal and altitudinal distributions of chemical $O_x$ production and ozone concentrations simulated for the different lightning schemes. Section 6 provides a novel approach to studying the

effects of flash frequency distribution on ozone. Section 7 presents the conclusions.

## 2 Model and data description

### 2.1 Climate-chemistry model

The model used is the UK Chemistry and Aerosols model (UKCA) coupled to the atmosphere-only version of the UK Met Office Unified Model version 8.4. The atmosphere component is the Global

Atmosphere 4.0 (GA4.0) as described by Walters et al. (2014). Tropospheric and stratospheric chemistry are modelled, although the focus of this study is the troposphere. The UKCA tropospheric scheme is described and evaluated by O'Connor et al. (2014) and the stratospheric scheme by Morgenstern et al. (2009). This combined *CheST* chemistry scheme has been used by Banerjee et al. (2014) in an earlier configuration of the Unified Model. There are 75 species with 285 reactions

considering the oxidation of methane, ethane, propane, and isoprene. Isoprene oxidation is included using the Mainz Isoprene Mechanism of Pöschl et al. (2000). Squire et al. (2015) gives a more detailed discussion of the isoprene scheme used here.

The model is run at horizontal resolution N96 (1.875° longitude by 1.25° latitude). The vertical dimension has 85 terrain-following hybrid-height levels distributed from the surface to 85 km. The

95 resolution is highest in the troposphere and lower stratosphere, with 65 levels up to $\sim 30$ km. The

model time step is 20 minutes with chemistry calculated on a 1 hour time step. The exception to this is for data used in section 6 where it was required that chemical reactions accurately coincide with time of emission and hence where the chemical time step was set to 20 minutes. The coupling is one-directional, applied only from the atmosphere to the chemistry scheme. This is so that the meteorology remains the same for all variations of the lightning scheme, and hence, differences in chemistry are solely due to differences in lightning $NO_x$.

The cloud parametrisation (Walters et al., 2014) uses the Met Office Unified Model's prognostic cloud fraction and prognostic condensate (PC2) scheme (Wilson et al., 2008a, b) along with modifications to the cloud erosion parametrisation described by Morcrette (2012). PC2 uses prognostic variables for water vapour, liquid and ice mixing ratios as well as for liquid, ice and total cloud fraction. The cloud ice variable includes snow, pristine ice and riming particles. Cloud fields can be modified by shortwave and longwave radiation, boundary layer processes, convection, precipitation, small-scale mixing, advection and pressure changes due to large-scale vertical motion. The convection scheme calculates increments to the prognostic liquid and ice water contents by detraining condensate from the convective plume, whilst the cloud fractions are updated using the non-uniform forcing method of Bushell et al. (2003).

Evaluation of the distribution of cloud depths and heights simulated by the Unified Model has been performed in the literature. For example, Klein et al. (2013) conclude that across a range of models, the most recent models improve the representation of clouds. They find that HadGEM2-A, a predecessor of the model used in this study, simulates cloud fractions of high and deep clouds in good agreement with the International Satellite Cloud Climatology Project (ISCCP) climatology. In addition, Hardiman et al. (2015) studied a version of the Unified Model which used the same cloud and convective parametrisations as used here. They found that over the tropical Pacific warm pool that high cloud of 10-16 km occurred too often compared to measurements by the CALIPSO satellite. This will bias a lightning parametrisation based on cloud-top height, over this region. Cloud ice content and updraught mass flux, which are used in the ice flux based lightning parametrisation presented in this study, are are not well constrained by observations and represent an uncertainty in the simulated lightning. However, these variables are fundamental components of the Non-Inductive Charging Mechanism and therefore it is appropriate to consider a parametrisation which includes such aspects.

Simulations for this study were set up as a time-slice experiment using sea surface temperature and sea ice climatologies based on 1995-2004 analyses Reynolds et al. (2007), and emissions and background lower boundary GHG concentrations, including methane, are representative of the year 2000. A one year spin-up for each run was discarded and the following year used for analysis.

## 2.2 Lightning NO emission schemes

The flash rate in the lightning scheme in UKCA is based on cloud-top height by Price and Rind (1992, 1993), with energy per flash and NO emission per joule as parameters drawn from Schumann and Huntrieser (2007). The equations used to parametrise lightning are:

$$F_l = 3.44 \times 10^{-5} H^{4.9} \tag{1}$$

$$F_o = 6.2 \times 10^{-4} H^{1.73}, \tag{2}$$

where $F$ is the total flash frequency (fl. $\min^{-1}$), $H$ is the cloud-top height (km) and subscripts l and o are for land and ocean, respectively (Price and Rind, 1992). A resolution scaling factor, as suggested by Price and Rind (1994), is used although it is small and equal to 1.09. An area scaling factor is also applied to each grid cell which consists of the area of the cell divided by the area of a cell at 30° latitude.

This lightning $NO_x$ scheme has been modified to have equal energy per cloud-to-ground and cloud-to-cloud flash based on recent literature (Ridley et al., 2005; Cooray et al., 2009; Ott et al., 2010). The energy of each flash is 1.2 GJ and NO production is $12.6 \times 10^{16}$ NO molecules $J^{-1}$ These correspond to 250 mol(NO) fl.$^{-1}$ which is within the estimate of emission in the review by Schumann and Huntrieser (2007). It also ensures that changes in flash rate produce a proportional change in emission independent of location since different locations can have different proportions of cloud-to-ground and cloud-to-cloud flashes. As a consequence, the distinction between cloud-to-ground and cloud-to-cloud has no effect on the distribution or magnitude of lightning $NO_x$ emissions in this study. The vertical emission distribution has been altered to use the recent prescribed distributions of Ott et al. (2010) and applied between the surface and cloud top. Whilst the Ott et al. (2010) approach is used for both lightning parametrisations, the resulting average global vertical distribution can vary because the two parametrisations distribute emissions in cells with different cloud top heights. This simulation with the cloud-top height approach will be referred to as CTH.

Two alternative simulations are also used within this study: 1) lightning emissions set to zero (ZERO), and 2) using the flash rate parametrisation of Finney et al. (2014) (ICEFLUX). The equations used by Finney et al. (2014) are:

$$f_l = 6.58 \times 10^{-7} \phi_{ice} \tag{3}$$

$$f_o = 9.08 \times 10^{-8} \phi_{ice}, \tag{4}$$

where $f_l$ and $f_o$ are the flash density (fl. $m^{-2} s^{-1}$) of land and ocean, respectively. $\phi_{ice}$ is the upward ice flux at 440 hPa and is formed using the following equation:

$$\phi_{ice} = \frac{q \times \Phi_{mass}}{c}, \tag{5}$$

where $q$ is specific cloud ice water content at 440 hPa (kg kg$^{-1}$), $\Phi$ is the updraught mass flux at 440 hPa (kg m$^{-2}$ s$^{-1}$) and $c$ is the fractional cloud cover at 440 hPa (m$^2$ m$^{-2}$). Upward ice flux was

set to zero for instances where $c < 0.01\,\mathrm{m}^2\mathrm{m}^{-2}$. Where no convective cloud top is diagnosed, the flash rate is set to zero.

Both the CTH and ICEFLUX parametrisations when implemented in UKCA produce flash rates corresponding to global annual NO emissions within the range estimated by Schumann and Huntrieser (2007) of 2-8 $\mathrm{TgN\,yr}^{-1}$. However, for this study we choose to have the same flash rate and global annual $\mathrm{NO_x}$ emissions for both schemes. A scaling factor was used for each parametrisation that results in the satellite estimated flash rate of $46\,\mathrm{fl.\,s}^{-1}$, as given by Cecil et al. (2014). The flash rate scaling factors needed for implementation in UKCA were 1.57 for the Price and Rind (1992) scheme and 1.11 for the Finney et al. (2014) scheme. The factor applied to the ice flux parametrisation is similar to that used in Finney et al. (2014), who used a scaling of 1.09. This is some evidence for the parametrisation's robustness since the studies use different atmospheric models, however, the scaling may vary in other models. Given that each parametrisation produces the same number of flashes each year and each flash has the same energy, a single value for NO production can be used. As above, a value of $12.6 \times 10^{16}\,\mathrm{NO\,molecules\,J}^{-1}$ was used for both schemes which results in a total annual emission of $5\,\mathrm{TgN\,yr}^{-1}$.

### 2.3 Lightning observations

The global lightning flash rate observations used are a combined climatology product of satellite observations from the Optical Transient Detector (OTD) and the Lightning Imaging Sensor (LIS). The OTD observed between $\pm 75^\circ$ latitude from 1995-2000 while LIS observed between $\pm 38^\circ$ from 2001-2015 and a slightly narrower latitude range between 1998-2001. The satellites were low earth-orbit satellites so did not observe everywhere simultaneously. LIS, for example, took around 99 days to twice sample the full diurnal cycle at each location on the globe. The specific product used here is referred to as the High Resolution Monthly Climatology (HRMC) which provides 12 monthly values on a $0.5^\circ$ horizontal resolution made up of all the measurements of OTD and LIS between May 1995 - December 2011. Cecil et al. (2014) provides a detailed description of the product using data for 1995-2010, which had been extended to 2011 when data was obtained for this study. The LIS/OTD climatology product was regridded to the resolution of the model ($1.875^\circ$ longitude by $1.25^\circ$ latitude) for comparison.

### 2.4 Ozone column and sonde observations

Two forms of ozone observations are used to compare and validate the model and lightning schemes. Firstly, a monthly climatology of tropospheric ozone column between $\pm 60^\circ$ latitude, inferred by the difference between two satellite instrument datasets (Ziemke et al., 2011). These are the total column ozone estimated by the Ozone Monitoring Instrument (OMI) and the stratospheric column ozone estimated by the Microwave Limb Sounder (MLS). The climatology uses data covering October 2004 to December 2010. The production of the tropospheric column ozone climatology by Ziemke

et al. (2011) uses the NCEP tropopause climatology so, for the purposes of evaluation, simulated ozone in this study is masked using the same tropopause. In Section 3.2, the simulated annual mean ozone column is regridded to the MLS/OMI grid of 5° by 5° and compared directly to the satellite climatology without sampling along the satellite track.

In an evaluation against ozone sondes with broad coverage across the globe, the MLS/OMI product generally simulated the annual cycle well (Ziemke et al., 2011). The annual mean tropospheric column ozone mixing ratio of the MLS/OMI product was found to have a root mean square error (RMSE) of 5.0 ppbv, and a correlation of 0.83, compared to all sonde measurements. The RMSE was lower and correlation higher (3.18 ppbv and 0.94) for sonde locations within the latitude range 25°S to 50°N.

Secondly, ozone sonde observations averaged into 4 latitude bands were used. The ozone sonde measurements are from the dataset described by Logan (1999) (representative of 1980–1993) and from sites described by Thompson et al. (2003) for which the data has since been extended to be representative of 1997–2011. The data consists of 48 stations, with 5, 15, 10 and 18 stations in the southern extratropics (90S-30S), southern tropics (30S-Equator), northern tropics (Equator-30N) and northern extratropics (30N-90N) respectively. In Section 3.2, the simulated annual ozone cycle is interpolated to the locations and pressure of the sonde measurements. The average of the interpolated points is then compared to the annual cycle of the sonde climatology without processing to sample the specific year or time of the sonde measurements. Both of these observational ozone datasets are the same as used in the Atmospheric Chemistry and Climate Model Intercomparison Project (ACCMIP) study by Young et al. (2013).

## 3 Comparison to observations

### 3.1 Global annual spatial and temporal lightning distributions

Using the combined OTD/LIS climatology allows extension of the evaluation made by Finney et al. (2014) which was over a smaller region. Figure 1 shows the satellite annual flash rate climatology alongside the annual flash rate estimated by UKCA using CTH and ICEFLUX. The annual flash rate simulated by UKCA is broadly representative of the decade around the year 2000 as it uses SST and sea ice climatologies for that period. A spatial correlation of 0.78 between the flash rate climatology estimated by ICEFLUX and the satellite climatology is an improvement upon the correlation of flash rates estimated by CTH which is 0.65. Furthermore, the root mean square error (RMSE) of the ICEFLUX climatology to the satellite data of 3.7 fl. $\mathrm{km}^{-2}\,\mathrm{yr}^{-1}$ is favourably reduced compared to the 6.0 fl. $\mathrm{km}^{-2}\,\mathrm{yr}^{-1}$ RMSE of the CTH climatology.

These results are similar to those found by Finney et al. (2014) who used offline ERA-Interim meteorology as the input to the parametrisation. Neither approach for simulating lightning achieves the observed ocean to land contrast despite using separate equations, and neither displays the large

peak flash rate in central Africa. The ICEFLUX approach over the ocean provides a contrast to the CTH approach by being an overestimate instead of an underestimate compared to the satellite lightning observations. While not achieving the magnitude of the observed Central African peak the ICEFLUX scheme does yield closer agreement over the American and Asian tropical regions.

Figure 2 shows comparisons of the monthly mean flash rates for 4 latitude bands. The ICEFLUX approach simulates lightning well in the extratropics with good temporal correlations with LIS/OTD in both hemispheres. The correlation of CTH with LIS/OTD is higher in the southern extratropics but this improvement compared to ICEFLUX is contrasted by much larger absolute errors. Correlations for both approaches are lowest in the southern tropics.

Figure 2B shows that CTH has very large root mean square errors during December to April in the southern tropics. A more detailed analysis (not shown) suggests that these errors are due to overestimation over South America. In the northern tropics the temporal correlation with LIS/OTD suggests CTH performs slightly better than the ICEFLUX approach, although Figure 2C shows that the CTH approach is not capturing the double peak characteristic of this latitude band. The ICEFLUX approach appears to simulate a double peak but it does not achieve the timing, which leads to a poor correlation. In the northern tropics, the more detailed analysis found that both schemes failed to match the observed magnitude of the August peak of Central America and the Southern US, nor the duration of the lightning peak over Northern Africa which lasts from June to September. The delay in the lightning peak that was apparent in annual cycles shown by Finney et al. (2014) over the tropics and subtropics is not so apparent here although there may be some delay in the southern tropics. The underestimation of ICEFLUX in the northern tropics and overestimation of CTH in the southern tropics found by Finney et al. (2014) is also found here.

Overall, the ICEFLUX approach reduces the errors in the annual cycles of lightning. This scheme improves the correlation between simulated and observed lightning compared to CTH scheme in the northern extratropics and southern tropics. It has a lower correlation in the northern tropics, where both approaches for simulating lightning have difficulties, and in the southern extratropics, where the magnitude of the bias is much reduced upon compared to the CTH approach.

To further understand how the schemes perform on a regional scale, the annual cycles of the simulated and observed lightning, for a selection of key regions, are shown in Figure 3A. A box showing each region is plotted on Figure 1. The regions of Figure 3 include many of the peak areas of lightning shown in Figure 1A or, in the case of Europe, are an area in which a higher density of measurement studies are undertaken including using ground-based lightning detectors.

Figure 3A shows the Central African peak lightning region where both parametrisations success-fully simulate the observed peak months of lightning in the LIS/OTD data. For the most part, both parametrisations produce similar flash rates. However the simulated flash rates generally underes-timate lightning compared to the observations. Interestingly, the ICEFLUX approach has a greater underestimation of the observed Spring lightning peak compared to the CTH approach. This sug-

gests that the input meteorology for the ICEFLUX scheme over the Central African region is less well simulated during this season, or that the ICEFLUX scheme does not capture some necessary aspect of thunderstorm activity during the season. Over the Indian region (Figure 3B), the two schemes substantially differ in their flash estimates. The ICEFLUX scheme achieves a much more realistic annual cycle than the CTH scheme. This suggests that aspects of charging during the Indian monsoon seasons may not be captured by the cloud-top height approach. Two regions in South America are shown in Figure 3 C and D. Both schemes capture the southern South American annual cycle of lightning flash rates well but both perform poorly in the northern region (the ICEFLUX approach results in a much lower bias). Biomass burning aerosols could be a key control on lightning activity in the region, as was shown by (Altaratz et al., 2010). The flash rate peak in the southern USA region is greatly underestimated by both schemes 3. The lack of difference between the two schemes suggests that it may not be the best study region for distinguishing which is a more successful parametrisation. Finally, over the southern European region, both schemes show an underestimation of flash rates compared to LIS/OTD, although the bias is less in the case of the ICEFLUX approach. The August peak in this region is not captured by either approach, which may relate to lightning activity over the Mediterranean Sea, given that both schemes also underestimate the annual flash rate over the Mediterranean Sea as shown in Figure 1.

The analysis of the annual cycle of flash rates in some key regions has shown that the ICEFLUX scheme is similar to or improves upon the simulated annual cycle by the CTH scheme when compared to the LIS/OTD satellite climatology. The exception is for the Central African peak in Spring. Any future studies of the Central African region could explore this difference further. Neither parametrisation captures the magnitude of flash rates over the southern USA or southern European regions. Given the high density of measurements in these regions it should be possible to study why this underestimation occurs in future studies. Finally, we suggest that one of the greatest sources of bias in the flash rate estimates by the CTH scheme are over northern South America. The ICEFLUX scheme reduces this bias but still does not capture the annual cycle. In southern South America both parametrisations reproduce the observed annual cycle of lightning. Therefore, we suggest that field campaigns comparing the southern and northern regions of South America would be particularly useful in improving the understanding of lightning processes and finding reasons for large-scale biases in models.

### 3.2 Global annual spatial and temporal ozone distributions

Ozone has an average lifetime in the troposphere of a few weeks and can be transported long distances during that time. It can therefore be challenging to identify the sources of measured ozone but we use two types of measurements here to analyse how lightning emissions influence ozone distribution. Satellite column ozone measurements provide estimates of effect on the annual hori-

zontal distribution of ozone whilst ozone sonde measurements demonstrate the altitudinal effect of lightning emissions on monthly varying ozone.

Comparisons with the MLS/OMI tropospheric column ozone climatology are made using Pearson correlations, RMSE and mean bias assessments. The model ozone is masked to the troposphere by applying the NCEP tropopause climatology to each month and regridding to the $5°$ by $5°$ horizontal resolution of the MLS/OMI climatology. Table 1 gives the annual results for the three simulations using CTH, ICEFLUX and ZERO lightning.

The inclusion of lightning emissions from either scheme has a large effect on the amount of ozone in the column as shown by the reduced mean bias and RMSE compared to the ZERO simulation, however, there is little difference between the two lightning schemes. There is a slightly larger mean bias with the ICEFLUX approach. To analyse the error in distribution without the bias present, an adjustment is made by subtracting the mean biases from the respective simulated ozone column distributions. Once this adjustment is made the ICEFLUX approach shows a slightly lower RMSE than the CTH approach (Table 1).

Figure 4 uses sonde measurements averaged over four latitudinal bands and taken at three pressure levels. The temporal correlations and mean biases of the model monthly means, interpolated to the same pressure and locations, against the sonde observations are shown.

Both lightning schemes show a reduction in mean bias compared to the ZERO run throughout all latitude bands and altitudes (Figure 4). The greatest impact of lightning is on the tropical, middle and upper troposphere. In these locations the ozone concentration simulated by the ICEFLUX scheme has a much better temporal correlation with sonde measurements than that simulated by the CTH scheme. The ICEFLUX approach has a larger bias than the CTH approach which is discussed further in the following paragraph.

Figure 5 shows the monthly ozone comparisons between sonde measurements and the model at $250\,$hPa and $500\,$hPa for the northern and southern tropics. It is clear that in the middle and upper troposphere the lightning scheme is important in achieving a reasonable magnitude of ozone. Both schemes still show an underestimate compared to observations all year round in the southern tropics and during spring in the northern tropics, but are within the variability of sonde measurements. Other aspects of simulated ozone chemistry or uncertainty in total global lightning emissions, which is $\pm 3\,$TgN on the 5 TgN used here, may contribute to this bias.

In Wild (2007) and Liaskos et al. (2015) the ozone burden and mean tropospheric column ozone respectively, scaled approximately linearly with increases in lightning emissions. Using the mean bias data in Table 1 we can calculate the mean increase in ozone column associated with each TgN emission from lightning. The average mean bias in ozone column of the ICEFLUX and CTH simulations is -3.0 DU, where as the mean bias of the ZERO simulation is -7.4 DU. Therefore, 5 TgN of lightning emissions has increased the mean ozone column by, on average, 4.4 DU. If we assume the effect of emissions is linear, these biases imply that the mean global effect of lightning on ozone

column is 0.9 DU TgN$^{-1}$. Changing lightning emissions to 8 TgN could increase the ozone column by 2.7 DU and result in a bias of less than 1 DU. Such bias potentially introduced by the uncertainty in total emissions or other aspects of the model is much greater than the difference in mean bias between the two lightning schemes given in Table 1. Therefore, the small difference in mean bias between the two lightning schemes does not necessarily imply greater accuracy, instead the correlation values between the model and sonde data (Figure 4) provide a more useful evaluation of parametrisation success.

In Figure 5 some features of the results from the simulations with lightning emissions stand out as being different from that in the ZERO run. These features occur as ozone peaks in April in the northern tropics (most notably at 500 hPa)(Figure 5D) and in October in the southern tropics (most notably at 250 hPa)(Figure 5A). The northern tropics peak in ozone improves the comparison to sondes at 500 hPa, if slightly underestimated. However, the 250 hPa April peak in Figure 5B does not appear in any of the model simulations. Potentially, the modelled advection is not transporting the lightning NO$_x$ emissions or ozone produced to high enough altitudes. An anomalous southern tropical peak in March in Figures 5A and C, particularly shown by CTH, is not shown in the sonde measurements, but this corresponds to a month where the CTH scheme especially is overestimating lightning, as seen in Figure 2. The ICEFLUX scheme is a much closer match to the lightning activity in the southern tropics in March and correspondingly the modelled ozone is less anomalous compared to the ozone sonde measurements in that month. The well modelled lightning activity in the southern tropics in October (Figure 2C) results in a correctly matched peak in the ozone sonde measurements at both pressure levels which does not occur in the ZERO run. From these comparisons to ozone sondes we conclude that the lightning emissions have impacts in particular months which include the months of peak ozone. Figure 2 shows that these are not necessarily the month of highest lightning activity in the region, but instead as the lightning activity builds in the region. It may be of particular use for field campaigns studying the chemical impact of lightning to focus on these months and, as discussed in Section 3.1, South America could provide a useful region in which to develop understanding of lightning activity and therefore also its impacts on tropospheric chemistry.

## 4   The influence of lightning on the global annual O$_x$ budget

The O$_x$ budget considers the production and loss of odd oxygen in the troposphere. Several studies have used O$_x$ budgets to study tropospheric ozone (Stevenson et al., 2006; Wu et al., 2007; Young et al., 2013; Banerjee et al., 2014). Here, the O$_x$ approach has particular use because it responds more directly to the emission of NO than O$_3$ which may form in outflows of storms and take several days to fully convert between O$_x$ species (Apel et al., 2015).

There are different definitions of $O_x$ family species and here we use a broad definition that includes $O_3$, $O(1D)$, $O(3P)$, $NO_2$ and several $NO_y$ species (Wu et al., 2007). The $O_x$ species and the different terms of the budget are illustrated in Figure 6. Of particular relevance to this study is the chemical production of $O_x$, the majority of which occurs through oxidation of NO to $NO_2$ by peroxy radicals. The ozone burden is considered along with the budget terms as it is the key species of interest and it makes up the majority of the $O_x$ burden.

The global annual $O_x$ budgets for CTH, ICEFLUX and ZERO are given in Table 2. These budget terms are for the troposphere. Here, the tropopause is defined at each model time step using a combined isentropic-dynamical approach based on temperature lapse rate and potential vorticity (Hoerling et al., 1993). Clearly, the ZERO simulation demonstrates the large control that lightning has on these budget terms with changes of around 20% in the ozone burden and chemical production and losses when lightning $NO_x$ emissions are removed (Table 2). The $O_x$ budget for the ZERO simulation shows that through reduced ozone production, there is reduced ozone burden and therefore chemical losses and deposition fluxes are reduced. The lifetime of ozone is given by the burden divided by the losses. Since the burden decreases more than the losses, the ozone lifetime reduces overall, although to a lesser extent than the burden and loss terms individually.

There is uncertainty in the global lightning $NO_x$ source of 2-8 TgN emissions (Schumann and Huntrieser, 2007), and there will be an associated uncertainty in the $O_x$ budgets. Using no lightning (ZERO) corresponds to a reduction of 5 TgN emissions over the year - less than the range of uncertainty in $LNO_x$. Therefore large changes in $O_x$ budget terms can be expected within the uncertainty range of the global lightning $NO_x$ emission total. In contrast, it would seem that for constant emissions of 5 TgN and a reasonable change in the flash rate distribution by using the ICEFLUX approach instead the CTH approach, there are only small differences in the global $O_x$ budget terms. The largest differences between the $O_x$ budgets of the ICEFLUX and CTH approaches are in the ozone burden and lifetime but these are only 2 %.

The $O_x$ budget discussed so far represents the troposphere, but if the whole atmospheric ozone burden is considered (Table 2) then it is apparent that there is an also a reduction in ozone in stratosphere which must be due to changes in the troposphere-stratosphere exchange of ozone. Previous studies have also found ozone produced from lightning is transported into the lower stratosphere (Grewe et al., 2002; Banerjee et al., 2014). In this study, we quantify the different transport between the two lightning schemes by considering differences in whole atmospheric ozone burden against differences in tropospheric ozone burden. The whole atmospheric ozone burden simulated with ICEFLUX approach is 13 Tg less than that simulated by the CTH approach. Given the tropospheric ozone burden simulated by the ICEFLUX approach is only 6 Tg less that that of the CTH approach, this means that the majority of the difference in ozone burden (∼55%) occurs in the stratosphere. On the other hand, the whole atmospheric ozone burden simulated in the ZERO run was 91 Tg less than that of the CTH approach. The tropospheric ozone burden was 62 Tg less so ac-

counts for around two thirds of the total difference in this case. The ICEFLUX approach has resulted
in less lightning emissions in the upper tropical troposphere and therefore less ozone is available in
the region to be transported into the stratosphere. We see that such a change in the lightning distribu-
tion, but maintaining the same level of total emissions, results in reduced net ozone production but
that much, and even the majority, of this reduction in ozone can occur in lower stratospheric ozone.

## 5 Differences in the zonal-altitudinal distributions of $O_x$ and $O_3$ between the two lightning schemes

In the previous section, it was demonstrated that the global tropospheric $O_x$ budget is affected prin-
cipally by the magnitude of emissions and not the location of emissions. This was achieved by
using the same total emissions but different distributions of lightning in the CTH and ICEFLUX
approaches (Figure 1), which simulate little difference in the global $O_x$ budget terms. This section
now considers changes in the zonal and altitudinal location of $O_x$ chemistry and ozone concentration
as a result of changes in the lightning emission distribution. The zonal-altitudinal net chemical $O_x$
production, as well as its components of gross production and loss, are shown in Figure 7A-C for
the CTH scheme as well as changes as a result of using ICEFLUX instead of CTH in Figure 7D-F.

The difference in net $O_x$ production when using the ICEFLUX scheme compared to the CTH
scheme is dominated by the change in gross production (Figure 7D and E). Figure 7E shows a shift
away from the tropical upper troposphere to the middle troposphere and the subtropics. There is over
a 10% reduction in the upper troposphere net production and 100% changes in the subtropics (Figure
7D). However, the high subtropical percentage change is principally due to small net production in
these regions. The changes in $O_x$ production result as a shift in emissions which happens by: 1)
reduced and more realistic lightning in the tropics (see Figure 8), and 2) decoupling of the vertical
and horizontal emissions distributions by not using cloud-top in both aspects (as is the case in CTH).
As described in section 2.2, the column $LNO_x$ is distributed up to the cloud-top, and this is how a
coupling exists between the horizontal $LNO_x$ distribution simulated by the CTH approach and the
height that $LNO_x$ emissions reach. This means that, by basing the horizontal lightning distribution on
cloud-top height and then distributing emissions to cloud top, $LNO_x$ is most effectively distributed to
higher altitudes. Hence, a lightning parametrisation for which the horizontal distribution is different
to that of cloud-top height will, to some extent, naturally distribute emissions at lower altitudes. This
is demonstrated best in Figure 7E which shows gross production in the northern tropics. Whilst both
lightning schemes have similar total lightning at these latitudes (shown in Figure 8), and therefore
similar column $O_x$ production, the gross $O_x$ production occurs less in the upper troposphere and
more in the middle troposphere when using the ICEFLUX scheme.

It is consistent with observations of lightning, that there is less lightning in the tropics than esti-
mated by CTH here. It is also consistent with current understanding that the most intense lightning

flash rates do not always occur in the highest clouds. We would therefore suggest that the change to
the net $O_x$ production of ICEFLUX is a more realistic representation of the distribution of production
than with CTH. The improved sonde correlations presented in section 3.2 support this conclusion.

Whilst $O_x$ gross production changes, mainly representing oxidation of NO to $NO_2$ by peroxy
radicals, show a close resemblance to the lightning NO emissions changes they are only part of the
picture with regard to changes in the distribution of ozone. This is because the lifetime of ozone
is much longer than the timescales for NO forming an equilibrium with $NO_2$. Furthermore, ozone
precursors are transported downwind of convection before they form ozone. The difference in $O_x$
production (Figure 7) between the two lightning schemes influences not only ozone locally but also
downwind where ozone is transported to.

Figure 9 presents the percentage changes in ozone distribution as a result of using the ICEFLUX
scheme instead of the CTH scheme. There is reduced tropical upper tropospheric ozone of up to
10% (Figure 9) due to reduced NO emission in that region. This results in less ozone transported
into the lower stratosphere under the ICEFLUX scheme compared to the CTH scheme. The lower
stratospheric ozone may also be lower due to less $NO_x$ being available for transport, and therefore
reduced chemical production in the stratosphere. Whilst ozone is lower in most of the lower strato-
sphere in the simulation with ICEFLUX the percentage changes are largest (up to 5%) nearer to the
tropopause.

In the middle and lower tropical troposphere there is also a reduction in ozone concentration (Fig-
ure 9) despite increased net $O_x$ production (Figure 7D). This is because there is less ozone produced
in the upper troposphere, and therefore there are lower ozone concentrations in the air transported
within the vertical circulation in the tropics. In the southern tropics, the net $O_x$ production increase
is due to reduced $O_x$ loss as a result of lower ozone concentrations in the region. Note that both
schemes experience the same meteorology because the chemistry is not coupled. The percentage
changes in ozone in the northern tropics are less than in the southern tropics (Figure 9). This is
likely to be in part due to offsetting through increased lightning emissions in the northern tropical
middle troposphere. Finally, the increased lightning emissions in the subtropics with the ICEFLUX
compared to the CTH scheme results in small changes in ozone throughout the extratropics.

It is worth noting that OH concentrations (not shown) respond in a similar manner to ozone
concentration with the change from the CTH to the ICEFLUX scheme. These changes are more
localised to emission changes but are still apparent in the lower stratosphere and extratropics. A
change from the CTH to ICEFLUX scheme results in only small changes in the methane lifetime as
a result of the changes in OH. Hence, in this setup we do not expect the ozone changes would be
greatly modified with the use of interactive methane.

Liaskos et al. (2015) identified that even with the same total global emissions, the magnitude and
distribution of radiative forcing resulting from lightning emissions is dependent on the method for
distributing the emissions horizontally and vertically. The changes in zonal-altitudinal distribution

discussed in this section show that these changes could be expected as a result of changes in ozone in the upper troposphere.

## 6 Frequency distributions of lightning and associated $O_x$ production

Lightning is a highly dynamic process. This section presents analysis of the frequency distribution of flash rates as a means to study the finer scale effects.

The CTH scheme simulates extremely low flash rates over the ocean. For instance, the maximum September oceanic flash rate using CTH was $1.1 \times 10^{-4} fl.km^{-2}20min^{-1}$ where as using ICEFLUX the maximum was over 100 times greater. This difference is not surprising given the difference in annual oceanic lightning activity shown in Figure 1. CTH tends to underestimate ocean lightning compared to satellite observations. The focus here will be on continental lightning. Other studies of frequency distribution in the literature have also focussed on continental locations so this work can be more directly compared to those.

Figure 10 shows the hourly continental flash rate frequency distribution for one model month (September). September was chosen as a month with a reasonable balance of lightning activity in between the hemispheres and where total lightning activity, and therefore emissions, was similar for the two lightning schemes.

When compared to the frequency distribution simulated by ICEFLUX, CTH has lower maximum flash rates, fewer occurrences of low flash rates and more occurrences of mid-range flash rates (Figure 10). Other studies have drawn similar conclusions regarding the frequency distributions of CTH when comparing to other parametrisations and lightning observations (Allen and Pickering, 2002; Wong et al., 2013; Finney et al., 2014). The ICEFLUX approach produces a similar distribution to that produced by the same scheme applied in the study by Finney et al. (2014). In that study the ICEFLUX frequency distribution had a fairly average distribution compared to four other lightning parametrisations with slightly more occurrences of low flash rates.

In Figure 10, the CTH frequency distribution displays some unusual periodic characteristics in the occurrence rate, most notably towards high flash frequencies. These features are also apparent in the cloud-resolving simulations presented in Wong et al. (2013). We suggest here that these features may arise due to discretised nature of the cloud-top height input variable.

The importance of the global flash rate frequency distribution to atmospheric chemistry frequency distributions is currently unknown but simplified model studies have suggested some key features:

– Compared to a set of observations over the US, a simulation using the CTH approach led to a greater ozone production efficiency due to the non-linear nature of ozone production and $NO_x$ (Allen and Pickering, 2002).

– Total ozone production increased approximately linearly up to 300 pptv of lightning $NO_x$ and then increased at a slower rate beyond that. This may be due to the ozone production approach-

ing the maximum possible for the given altitude, solar zenith angle and $HO_x$ concentration (DeCaria et al., 2005).

In the following analysis we consider $O_x$ production rather than ozone production because it exhibits a more immediate response to NO emission. This is important given the difficulty and errors associated with tracking ozone production associated with each emission source in a global model. However, there are some comparable results which we will compare to the previous findings above, as well as new insights into the consequences of different frequency distributions and lightning parametrisations.

Figure 11 presents two metrics of the gross column chemical $O_x$ production resulting from continental lightning in each of the frequency bins of Figure 10. The metrics are: A) the mean column $O_x$ production, and B) the mean $O_x$ production per flash. Each flash corresponds to $250\,mol(NO)$ emission so the $O_x$ production per mole of emission can easily be inferred from the $O_x$ production per flash. $O_x$ production resulting from lightning is calculated as the difference between the model run with lightning and the model run with no lightning, using the grid cells from the no lightning run that correspond to the cells used in each bin for the relevant lightning parametrisation. This means that this work is focussing on the *initial* $O_x$ production occurring in the 20 minute time step in which emissions are produced. This initial $O_x$ production has been calculated to be approximately 15% of total $O_x$ production associated with lightning for both parametrisations. The calculation was made as the difference between the total $O_x$ production resulting from lightning in the sampled grid cells and the total $O_x$ production resulting from lightning over the whole globe in all time steps. The remaining 85% of production must occur after the initial time step and be a result of advected emissions or changes to the large-scale distributions of constituents such as ozone or OH as discussed in section 5.

The mean column $O_x$ production in Figure 11A shows, as expected, that increasing flash rate (i.e. more NO emissions in a cell) results in increased column $O_x$ production. The higher extreme flash rates of ICEFLUX compared to CTH result in greater column $O_x$ productions as a result of individual occurrences. A linear increase in $O_x$ production is apparent up to approximately $0.02\,fl.\,km^{-2}\,20min^{-1}$ at which point the two schemes produce 1 to $1.5\,kg\,km^{-2}\,20min^{-1}$ of $O_x$. Beyond this point, the $O_x$ production simulated by the ICEFLUX approach increases still linearly but with a shallower gradient. The ICEFLUX scheme produces less $O_x$ for a given flash rate than the CTH scheme at higher flash rates but more at lower flash rates (Figure 11A). This is due to emissions from high flash rates in ICEFLUX not necessarily being distributed to such high altitudes as with CTH. At the higher altitudes that emissions reach when using the CTH scheme, $NO_x$ has a greater ozone production efficiency, as discussed in section 5. Conversely, in the ICEFLUX scheme, lower flash rates can occur in relatively deeper cloud so in these there can be greater $O_x$ production efficiency compared to the CTH scheme because the CTH scheme will always place these low flash

rates at lower altitudes. On larger scales, whilst high extreme flash rates produce more $O_x$, they occur relatively infrequently so do not greatly affect the global $O_x$ budget.

Figure 11B shows the mean column $O_x$ production per flash for each flash rate bin. It is derived by dividing the data in Figure 11A by the mid-point flash rate of each bin. Whilst Figure 11A shows that lower flash rates produce less $O_x$, they do produce $O_x$ more efficiently than higher flash rates. Flash rates of $0.0005\,\mathrm{fl.\,km^{-2}\,20\,min^{-1}}$ produce $\sim 10$ times more $O_x$ per flash than flash rates of $0.05\,\mathrm{fl.\,km^{-2}\,20min^{-1}}$. This suggests that as the NO increases, $NO_x$ cycling and therefore ozone production decreases in efficiency. This is likely a result of peroxy radical availability and VOC abundance limiting the rate of $NO_x$ cycling. Evidence for such control of VOC precursors on ozone production in US thunderstorms has been presented by Barth et al. (2012).

ICEFLUX displays the greatest contrast in efficiency between high and low flash rates of the two parametrisations (Figure 11B). As with the column mean production, because the CTH scheme places the most emissions in the highest cloud tops it is more efficient at producing $O_x$ at higher flash rates but the ICEFLUX scheme is more so at lower flash rates. Using the NO production per flash of $250\,\mathrm{mol(NO)\,fl.^{-1}}$ stated in Section 2.2, the range of initial $O_x$ production per mol of emission is $25\,\mathrm{mol(O_x)\,mol^{-1}(NO)}$ at low flash rates for ICEFLUX to less than $2\,\mathrm{mol(O_x)\,mol^{-1}(NO)}$ for the highest flash rates in the ICEFLUX scheme (Figure 11B).

In summary, we find similarly to Allen and Pickering (2002) that $O_x$ production becomes less efficient at higher flash rates. It is important to consider that in our case the higher flash rates are less efficient at the point of emission - the emissions may go on to produce $O_x$ elsewhere following advection. Also, similarly to DeCaria et al. (2005), we find that the mean column $O_x$ production increases linearly up to a point, in our case $0.02\,\mathrm{fl.\,km^{-2}\,20min^{-1}}$, then increases at a slower, but still linear rate beyond that. New insights provided through the use of a global model are:

- Both lightning schemes produce about 15% of the $O_x$ associated with lightning in the first 20 minutes after the time of emission

- For the CTH approach, oceanic flash rates are so low that associated $O_x$ production at the time of emission is negligible for the global production

- Because CTH places the most emissions in the highest clouds (where ozone production efficiency is greater), more $O_x$ is produced by the CTH scheme than ICEFLUX at high flash rates, but ICEFLUX produces more at low flash rates

- Initial $O_x$ production per flash is approximately 10 times greater for low flash rates than high-end flash rates

These findings regarding the $O_x$ production per flash provide a useful metric to evaluate lightning parametrisations with observations. Several differences between the CTH and ICEFLUX scheme suggest further study is needed to determine the true nature of $O_x$ production. For instance, the

almost negligible proportion of $O_x$ production that will occur over the ocean when using the CTH scheme due to very low flash rates would benefit from oceanic measurements of ozone and $NO_x$ in the vicinity of storms. This study has analysed the $O_x$ production occurring in the first 20 minutes, but further $O_x$ production can occur over longer time periods. An extension of the work here could be to run idealised experiments of pulse lightning emissions in a global model to see how the $O_x$ and ozone production develop with time and hence, assess the lag between NO emission and ozone production.

## 7  Conclusions

A new lightning parametrisation based on upward cloud ice flux, developed by Finney et al. (2014), has been implemented in a chemistry-climate model (UKCA) for the first time. It is a physically based parametrisation closely linked to the Non-Inductive Charging Mechanism of thunderstorms. The horizontal distribution and annual cycle of flash rates as calculated through the new ice flux approach and the commonly-used, cloud-top height approach were compared to the LIS/OTD satellite climatology. The ice flux approach is shown to generally improve upon the performance of the cloud-top height approach. Of particular importance is the realistic representation of the zonal distribution of lightning using the ice flux approach, whereas the cloud-top height approach overestimates the amount of tropical lightning and underestimates extra-tropical lightning.

The ice flux approach greatly improves upon the cloud-top height approach in UKCA with regards to the temporal correlation to the observed annual cycle of ozone in the middle and upper tropical troposphere. Through considering a simulation without emissions and the simulated annual cycle of lightning, it is clear that the ice flux approach reduces the biases in ozone in months where the cloud-top height approach has the largest errors in simulating lightning.

The zonal flash rate distribution when using the ice flux approach instead of the cloud-top height approach results in a shift of $O_x$ production away from the upper tropical troposphere. As a consequence there is a 5-10% reduction in upper tropical tropospheric ozone concentration along with smaller reductions in the lower stratosphere and small increases in the extratropical troposphere. These changes in ozone concentration are a result of the change in distribution of lightning emissions only, the total global emissions are the same for both schemes. We conclude that biases in zonal lightning distribution of the cloud-top height scheme increase ozone in the upper tropical troposphere and, as demonstrated by comparison to ozone sondes, this reduces the correlation to observations in ozone annual cycle in this region.

Analysis of the continental flash rate frequency distribution shows the cloud-top height approach has lower high-end extreme flash rates, more frequent mid-range flash rates and less frequent low-end flash rates, compared to the frequency distribution using the ice flux approach. Such features simulated by the cloud-top height approach have been found in comparisons to the observed fre-

quency distribution over the US and this current evidence suggests such a frequency distribution is unrealistic. We apply a novel analysis to determine the impact of the differences in flash rate frequency distribution on the initial $O_x$ production resulting from lightning emissions. As expected, the higher the flash rate, the more $O_x$ is initially produced. However, the $O_x$ production efficiency reduces for higher flash rates; lower flash rates initially produce approximately 10 times as much $O_x$ as higher flash rates. Further study is warranted to determine how emissions produce ozone downstream of a storm in complex chemistry models, but the result here is relevant to aircraft campaigns measuring $NO_x$ and ozone near to the thunderstorms. It would be useful to study such measurements to determine if less intense storms exhibit such a difference in $O_x$ production efficiency.

The global lightning parametrisation of Finney et al. (2014) using upward cloud ice flux has proven to be robust at simulating present-day annual distributions of lightning and tropospheric ozone. The reduced ozone in the upper tropical troposphere could be important for the understanding of ozone radiative forcing. In addition, the differences in the frequency distribution when using different lightning schemes is shown to affect the chemical $O_x$ production. The parametrisation is appropriate for testing in other chemistry transport and chemistry-climate models where it will be important to determine how the parametrisation behaves using different convective schemes. Furthermore, this new parametrisation offers an opportunity to diversify the estimates of the sensitivity of lightning to climate change which will be the focus of future work.

## 8 Author contribution

DLF, RMD, OW and NLA designed the experiments and interpreted the results. DLF performed the analysis. DLF and NLA developed the code and ran simulations. DLF prepared the manuscript with contributions from all co-authors.

*Acknowledgements.* This work has been supported by a Natural Environment Research Council grant NE/K500835/1. We thank the TRMM satellite team for access to the Lightning Imaging Sensor products. Thanks to Paul Young for providing and assisting with use of the ozone column and sonde observations, and Jonathan Wilkinson for guidance regarding implementation of the lightning parametrisation based on ice flux in the Met Office Unified Model. Finally, we thank the two anonymous reviewers who greatly helped to improve the manuscript.

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

**Table 1.** Spatial comparisons of correlation, errors and bias of annual tropospheric ozone column between model runs and the MLS/OMI satellite climatology product over the range $\pm 60°$. Adjusted root mean square error (RMSE) refers to the RMSE following the subtraction of the mean bias from the field.

| Run | r | RMSE (DU) | Mean bias (DU) | adjusted RMSE (DU) |
|---|---|---|---|---|
| CTH | 0.82 | 5.5 | -2.8 | 4.1 |
| ICEFLUX | 0.84 | 5.7 | -3.2 | 3.9 |
| ZERO | 0.83 | 10.7 | -7.4 | 4.6 |

**Table 2.** Global annual tropospheric $O_x$ budget terms for the year 2000 for three different simulations: CTH, ICEFLUX and ZERO. All terms in $Tg\,yr^{-1}$ except Burden which is in Tg and lifetime which is in days. The percentage difference with respect to the CTH budget is shown in brackets. In addition to the tropospheric budget terms, the whole atmospheric ozone burden is also included.

| | CTH | ICEFLUX | ZERO |
|---|---|---|---|
| Chem. prod. | 4472 | 4443 (-1%) | 3638 (-19%) |
| Chem. loss | 3848 | 3821 (-1%) | 3115 (-19%) |
| Net chem. prod. | 624 | 622 (0%) | 522 (-16%) |
| Deposition | 1006 | 1006 (0%) | 899 (-11%) |
| Strat. influx* | 382 | 384 (0%) | 376 (-2%) |
| Trop. $O_3$ burden | 267 | 261 (-2%) | 205 (-23%) |
| Whole atm. $O_3$ burden | 3253 | 3240 | 3162 |
| $\tau_{O_3}$ | 19.8 | 19.5 (-2%) | 18.4 (-7%) |

* Stratospheric influx is inferred to complete the $O_x$ budget through balancing the chemical loss and production and deposition.

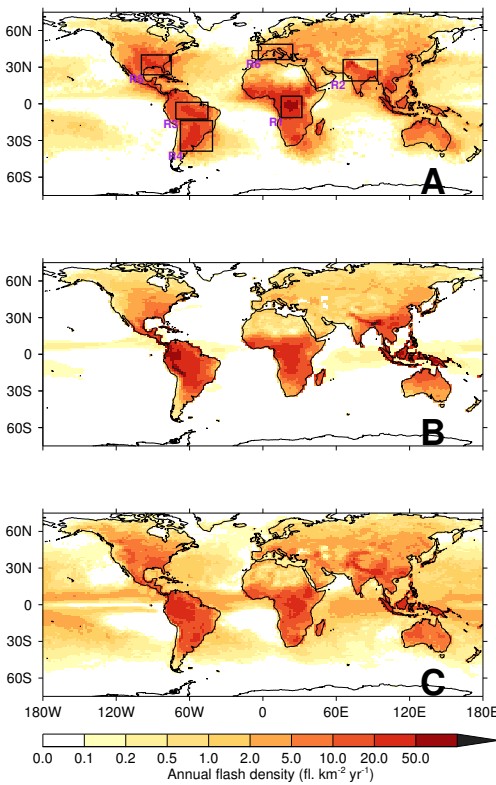

**Figure 1.** Annual flash rates from (A) a combined climatology from LIS/OTD satellite observations spanning 1995-2011, (B) the CTH scheme using the year 2000 of UKCA output and (C) the ICEFLUX scheme using the year 2000 of UKCA output. The horizontal resolution of the climatology product has been degraded to match that of the model which is $1.875°$ longitude by $1.25°$ latitude. Boxes for the regions R1–R6 correspond to regions of interest for which the annual cycles are shown in Figure 3.

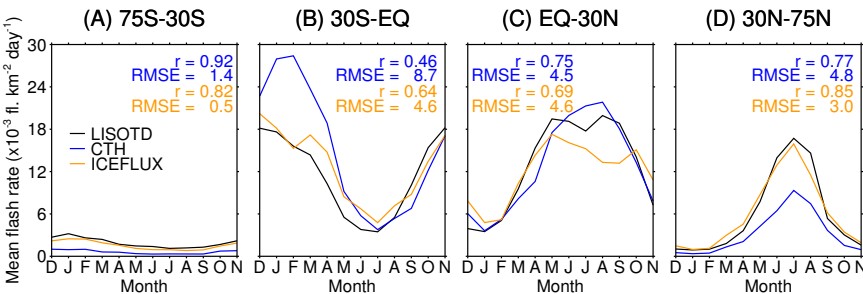

**Figure 2.** Mean monthly flash rate averaged over four latitudinal bands for the two different schemes for 2000 and the LIS/OTD climatology spanning 1995-2011. The points use one year of UKCA model output and a combined climatology from LIS/OTD satellite observations spanning 1995-2011. Also given are the temporal correlations (r) between the CTH scheme (blue) and LIS/OTD and between ICEFLUX (orange) and LIS/OTD. The corresponding root mean square errors (RMSE) are given in units of $10^{-3}$ fl. $km^{-2}yr^{-1}$.

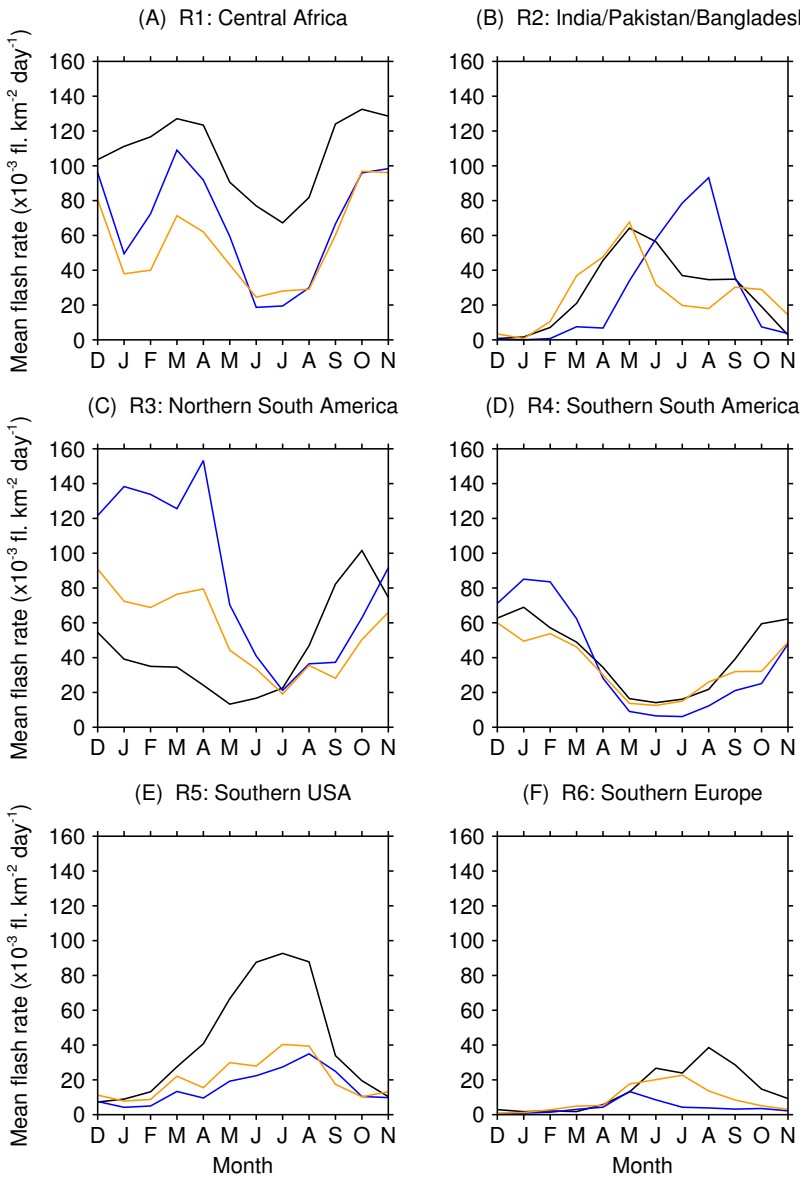

**Figure 3.** Mean monthly flash rate averaged over six regions (R1–R6) for the two different schemes for year 2000 and the LIS/OTD climatology spanning 1995-2011. Lines represent the lightning simulated using the CTH approach (blue) and the ICEFLUX approach (orange), and the LIS/OTD observed climatology (black). Regions R1-R6 are shown as boxes on Figure 1.

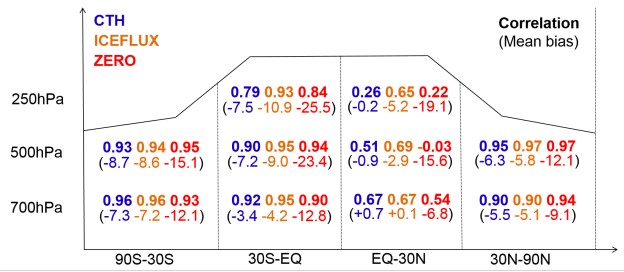

**Figure 4.** Temporal correlations and mean biases of the annual cycle of modelled ozone in UKCA over the year 2000 compared to a climatology of ozone sonde measurements averaged over 1980-1993 and 1997-2011. The simulated ozone data was interpolated to the location and pressure level of the sonde measurements. The sonde and modelled ozone were then averaged into 4 latitude bands which correspond to the bands used in Figure 2.

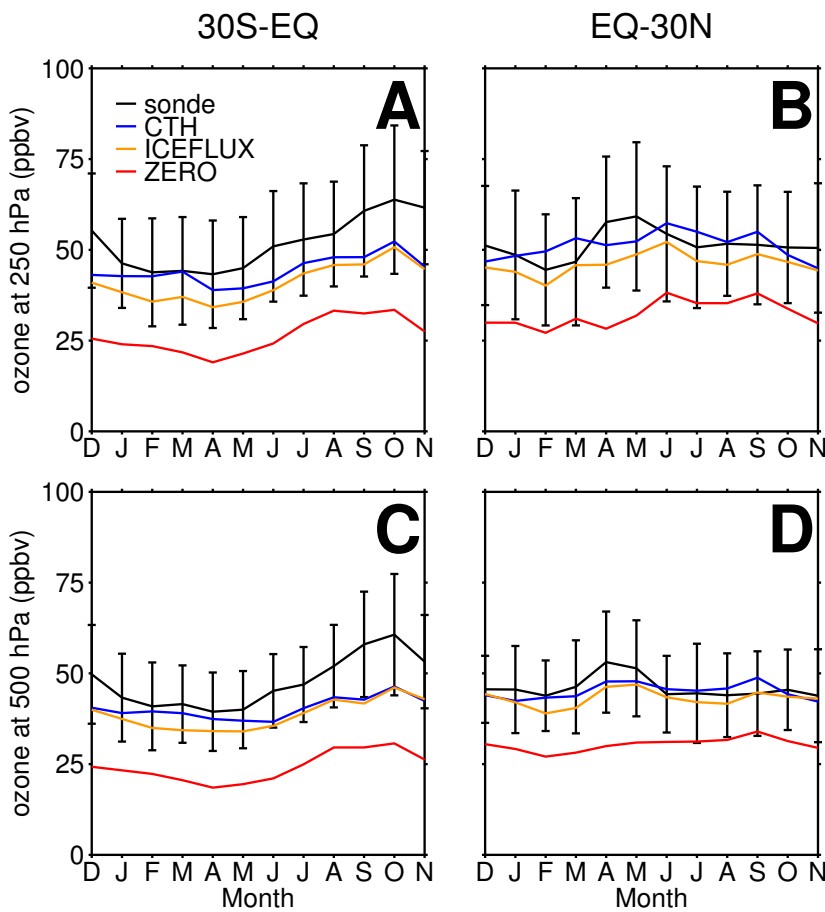

**Figure 5.** Middle and upper tropospheric UKCA simulated ozone concentration for the year 2000 compared to a climatology of sonde measurements averaged over 1980-1993 and 1997-2011. These cycles correspond to the 500 hPa and 250 hPa correlations for 30S-EQ and EQ-30N in Figure 4. The vertical black bars show the average interannual standard deviation for each group of stations.

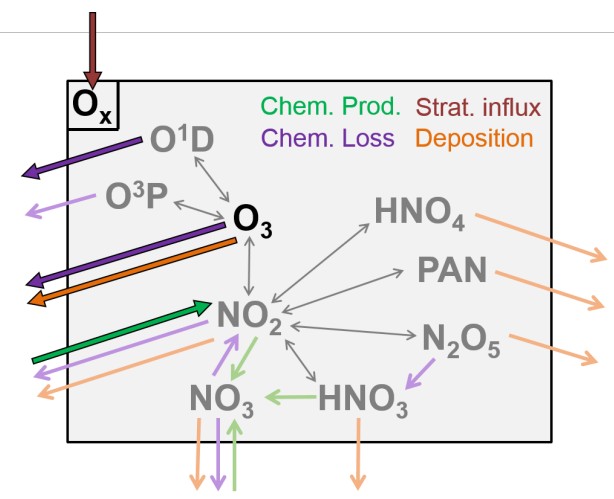

**Figure 6.** The UKCA definition of $O_x$ species and the $O_x$ budget. Major contributors are shown in bright colours and black outlines, minor contributors in pale colours. Grey arrows are reactions between $O_x$ species and therefore result in no production or loss. The stratospheric influx is not determined for individual species. Instead the total $O_x$ influx is inferred to balance the production and loss terms. The burden and stratospheric influx of $O_x$ are dominated by the burden and stratospheric influx of $O_3$.

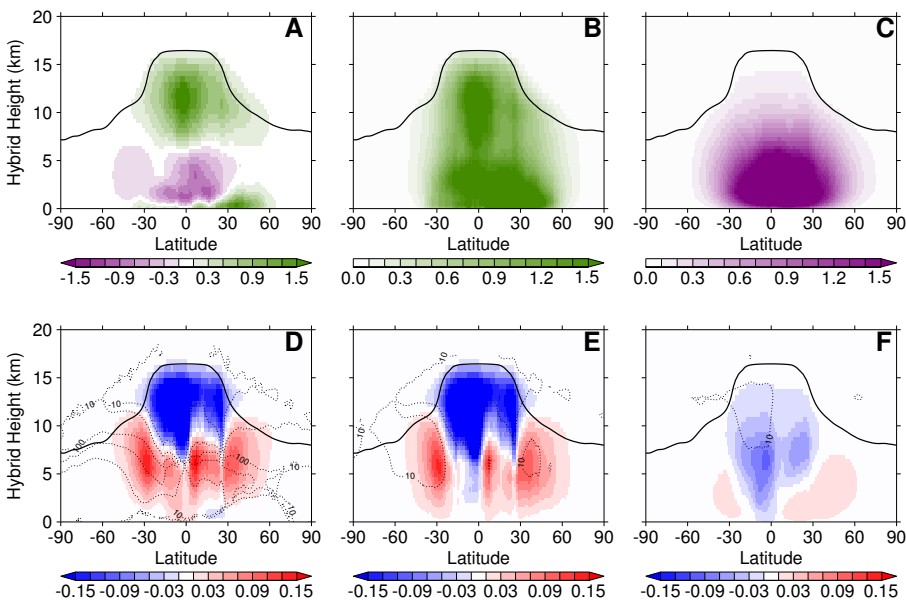

**Figure 7.** Annual total zonal-altitudinal distributions of $O_x$ reaction fluxes for CTH for the year 2000. These fluxes are A) Net production, B) gross production, and C) gross loss of $O_x$. The respective differences between simulations using the ICEFLUX scheme and the CTH scheme are shown in D-F. All units are $Tg(O_3)$. Values are annual and meridional totals. The solid line is the annual mean tropopause and dashed lines contour 10% and 100% changes. The $O_x$ fluxes were masked with the model tropopause every time step.

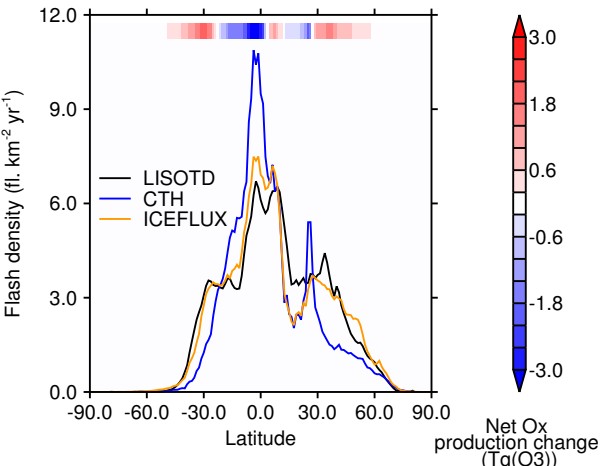

**Figure 8.** Zonal mean lightning flash rate from the LIS/OTD climatology and as modelled by CTH and ICE-FLUX. The zonal changes in net tropospheric column $O_x$ production (ICEFLUX-CTH) are shown by the colour bar. The units of $O_x$ are expressed as a mass of ozone.

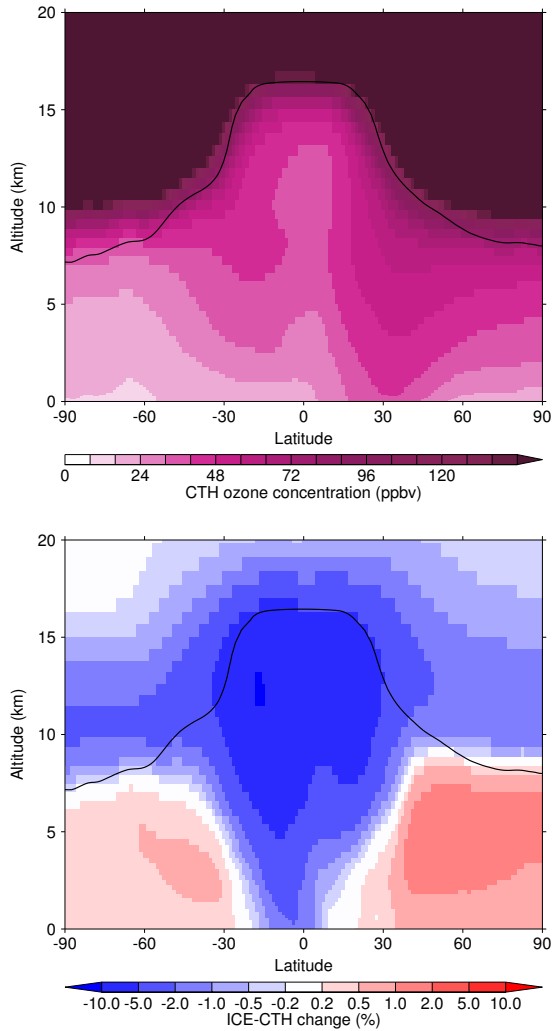

**Figure 9.** Annual mean distribution of ozone concentration modelled using the CTH approach, and the percentage difference between ICEFLUX and CTH simulated ozone concentration. The solid line shows the mean annual tropopause as diagnosed using the modelled meteorology.

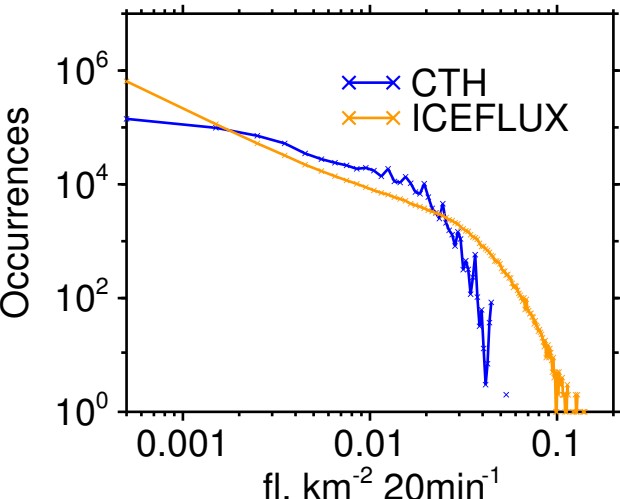

**Figure 10.** Frequency distribution of continental lightning flash rates using all time steps, for one month (September 2000) as modelled by the CTH and ICEFLUX schemes. The binsize used is 0.001 $\mathrm{fl.\,km^{-2}20min^{-1}}$ with crosses placed at the centre value of each bin.

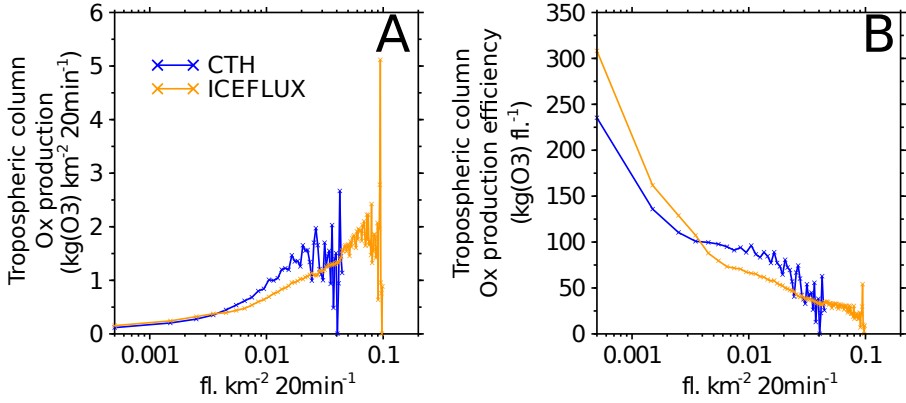

**Figure 11.** Two metrics of intial gross column $O_x$ production as a result of continental lightning simulated by the CTH and ICEFLUX schemes. The cells used in each bin correspond to those used in Figure 10. The metrics are A) mean column $O_x$ production in each bin, and B) mean column $O_x$ production per flash in each bin. The $O_x$ production resulting from lightning was determined by subtracting the column $O_x$ production in the no lightning run from the each lightning parametrisation for the corresponding cells. To reduce noisiness, only data is only plotted up to the highest bin of each parametrisation where there are at least two occurrences in Figure 10. The units of $O_x$ are expressed as a mass of ozone.