# Peer review of "The impact of lightning on tropospheric ozone chemistry using a new global lightning parametrisation"

_Atmospheric Chemistry and Physics, 2016_

## Referee Comment (RC1) · Anonymous Referee #1 · 14 Mar 2016

In this manuscript the authors follow up the Finney et al. (2014, ACP) paper that introduced a new scheme for estimating lighting flash rates using the upward flux of cloud ice in thunderstorms. Here this scheme is employed in the UK Chemistry and Aerosols model, which is driven by the meteorological fields from the UK Met Office Unified Model. The ICEFLUX scheme is shown to be superior to the cloud-top height scheme in predicting flash rate distributions when compared with the OTD/LIS climatology. The results also show that the ozone distributions are also improved when using the ICE-FLUX scheme, when compared with OMI/MLS ozone columns and with ozonesonde observations. The manuscript is well written and well organized. There are a few additional analyses that could be added, which I outline below, but these should involve only a minor effort. The manuscript should be published after attention is given to these items.

[Figure]

Significant Comments: Lines 98-99: Are there separate variables for cloud ice and precipitation-size ice (snow and graupel)? If so, please be more specific here.

Line 171: Are there any biases in this tropospheric ozone product relative to sondes or other satellite observations?

Lines 219-220: The correlation is also not improved with ICEFLUX in the southern extratropics.

Lines 236-237: It would benefit the paper if these statistics were presented for the latitude bands.

Line 300: It is not clear how you are defining the tropopause. How are the 380K surface and the 2 PVU surface combined?

Line 436: I am not sure what is the significance of this Ox production in the first 20 minutes. Isn't it primarily just the production of NO2 as it comes into equilibrium with NO in the atmosphere after flashes occur? Very little ozone production is going to be produced in 20 minutes.

Table 2: What are the percentages? Are they the percentage changes from the CTH scheme? MOre meaningful might be to show the percentage changes of CTH and ICEFLUX from ZERO.

Figure 5: There should be stratospheric influx of NO2 and other NOy species.

Minor Comments: Line 21: comparison of models

Line 183: define ACCMIP

Line 267: .....instead the correlation values between the model and the sonde data (Figure 3) provide a more......

Line 302: ....production and losses when lightning is added (Table 2).

Line 446: The increase is linear up to approximately 0.006 fl km-2 min-1 and then

becomes steeper up to o,02 fl km-2 min-1 at which....

---

## Referee Comment (RC2) · Anonymous Referee #2 · 14 Mar 2016

Summary

This article presents results from a global chemistry-climate model to show how ozone distributions and production rates change when the authors' new lightning parameterization (which is based on upward cloud ice flux) is used compared to the previous lightning parameterization that is based on cloud top height and implemented in several chemistry-climate and chemistry-weather models. Lightning produces nitrogen oxides (NOx) in the middle and upper troposphere and therefore impacts troposphere ozone (O3) especially in the upper troposphere. One weakness in our ability to predict the production of NOx from lightning is to be able to predict the lightning flash rate. Finney et al. developed a new lightning flash rate parameterization for global models, and showed that it compares well with observations (in both a previous paper using a model driven by reanalysis data and in this paper using a climate model). In this paper,

the authors describe how this new flash rate parameterization, which tends to predict fewer flashes in the tropics and more flashes in the continental extratropics, affects tropospheric ozone distributions and production rates. Because many models use the previous parameterization, it is important to publish this evaluation of the new lightning parameterization on tropospheric ozone chemistry.

The paper does a nice job of evaluating the model results to observations, comparing results between different sensitivity simulations, and showing that the ozone production efficiency is less for regions with high flash rates relative to regions with low flash rates. Although they make good recommendations, e.g. focusing field campaigns during particular months, I think there could be some additional analysis work on specific regions that can also guide future field campaigns and regional-scale model simulations. There are parts of the paper that need to be clarified. I would recommend that the authors pay attention to paragraph construction, being sure the paragraph begins with a topic sentence and is followed by supporting information linking sentences from what the reader knows to the new information (e.g. Schultz, Eloquent Science, 2009). I recommend minor revisions before publication.

Specific Comments

Title. To convey what kind of parameterization, I would suggest modifying the title to say, "using a new lightning parameterization".

L. 126. How does the lightning-NOx scheme differentiate between cloud-to-ground (CG) and intracloud (IC) lightning? Does it need to make this difference if the production of NO per flash is the same for CG and IC flashes and the vertical distribution of NO sources is the same for CG and IC flashes?

Section 3. Both lightning flash rate schemes depend on how well the model predicts cloud top height or ice mass flux. Has there been an attempt to evaluate these parameters (or a proxy) from the chemistry-climate model to a climatology of reanalysis data?

Section 3.1 has a nice analysis of lightning flash rates in different latitude bands, and remarks upon differences in continental regions. I wonder if an additional figure showing how the model performs for different continents could be included and discussed. For example, showing the annual cycle of North American, South American, African, India and East Asia, and Australia (and maybe tropical oceanic region) lightning should give peaks at different times of the year. This type of figure would be a natural follow on to Figure 1 because the eye is drawn to each of these regions when viewing Figure 1.

L. 252. While both lightning-NOx schemes show a general underestimate of ozone in the middle and upper troposphere of the tropics, they are both within the variability of the observations (while no lightning-NOx is outside that variability). In fact, the northern tropics appears to have quite good agreement. If you want to point out the underestimation, restrict the comment to the southern tropics. Second, what is the variability in the model results?

L. 284. I like the conclusion from the analysis that point to April and October as specific months to focus field campaigns. However, aircraft field campaigns can only cover a region (and not a latitude band). Can you recommend where field campaigns should focus? A similar analysis of continental regions would be helpful.

L. 297. I assume that the major Ox production is through oxidation of NO by peroxy radicals. This should be clarified to avoid confusion with NO + O3 producing NO2. It is curious that Table 2 discusses production and loss rates of Ox, but burdens of O3. I assume that is because O3 is the dominant Ox species (although it is of equal size to NO2 and O(1D) in Figure 5). It would be good to clarify in this paragraph why you discuss O3 burdens juxtaposed with the Ox production and loss rates discussion.

L. 303-304. Perhaps the characterization of the ZERO case could be revised. I think it should be described as the following. There is less production of Ox (or O3) without lightning-NOx emissions, resulting in a smaller O3 burden and therefore reduced

[Figure]

Ox losses and shorter O3 lifetime. Can anything be said about linear or non-linear responses? For example, it seems that the lifetime decrease is less than the Ox loss rate decrease, and both are less than the decrease in Ox production.

Section 4. In addition to the comparison of the ZERO case with the two other cases, there should be a statement pointing out similarities between ICEFLUX and CTH, including the point at the beginning of Section 5.

L. 341-344. I think it would be helpful to the reader to repeat how the NO lightning emissions are placed vertically for each scheme. It is also not clear to me how the horizontal distribution affects the vertical distribution. My interpretation is that ICEFLUX predicts lower lightning-NOx emissions in the tropics based on the storm parameters and more in the extratropics. While the magnitude of NO emissions is less in the ICEFLUX scheme for the tropics, those emissions are still distributed according to the Ott et al. (2010) curves to cloud top height (lines 126-128; and cloud top height should be the same in the two simulations). However, I think the authors are trying to say that the ICEFLUX scheme produces a lot of lightning-NO emissions in storms with lower cloud tops. There is also the point that because the CTH scheme has greater NO emissions in taller clouds, there is a substantial difference in where the NO emissions are found vertically. I think this could easily be supported by a plot of lightning-NO emissions versus cloud top height for different latitude bands.

L. 369-375. Could this be clarified? It was already established that a reduction in Ox production decreased O3 mixing ratios and therefore Ox loss rates (Section 4). However, in these lines it says there is an increase in Ox production in the middle and lower troposphere but a reduction of O3 concentrations, when comparing ICEFLUX and CTH results. Is Ox partitioned differently, meaning there is more HNO3 that can be removed? What loss process dominates (O3 chemical loss or Ox wet deposition)?

L. 405-410. I was surprised that the ICEFLUX lightning flash rate frequency distribution was not discussed. Also, although it is not the point of section 5, I wonder if it would be

useful to include LIS/OTD frequency distribution in Figure 9.

L. 460. It is an interesting finding that Ox production efficiency is less for higher flash rates (at least initial Ox production). Could the authors speculate why this would happen? Or suggest analysis that could be done in order to explain why. I would imagine the HO2 and RO2 abundance might play a role. Are there connections between flash rate and location to VOC sources? For example, Barth et al. (2012) showed more O3 produced from storms occurring over VOC-rich regions (e.g. southeast U.S.).

L. 465. How did the authors translate the Ox production efficiencies to Ox produced per mole of NO?

L. 477. Here, the authors argue that more Ox is produced by the CTH scheme because NOx has a longer lifetime at higher altitudes. However, the analysis is for the initial Ox production ("at the time of emission")? How does the NOx lifetime affect the Ox production shown in Figure 10, which is "at the time of emission"?

Technical Comments

L. 9 Insert "NO" before emission.

L. 17 Replace "-" with ";"

L. 16-18 I suggest adding a caveat that more ozone production can subsequently occur from the high flash rate regions.

L. 21-22 Change to "for comparison between models and observations . . .".

L. 27 NO2 lifetime may be shorter in the upper troposphere because its photolysis rate is greater. I think it would be better to rewrite the sentence to say NOx lifetime is longer in the upper troposphere (rather than the individual species).

L. 51 Could a reference be cited supporting that the upper troposphere is the region with most efficient ozone production?

L. 53 Please delete "simplified". I find cloud chemistry models to be rather complex.

L. 63-64 It would be better written as, "... of low flash rates, which are unrealistic compared to observed flash rates. This results in low NOx concentrations and greater ozone production efficiency ...."

L. 86 Please add more information about the chemistry represented in the model. Is it the "standard troposphere" chemistry or does it have the added isoprene chemistry, both described in O'Conner et al. (2014)? I suggest including number of species, stating it describes methane, ethane, and propane (and maybe isoprene) hydrocarbon chemistry.

L. 147-151 Could this be rewritten? It appears that only lightning flash rates are scaled to obtain a global values of 46 fl/s, because the NO production per energy is the same for both cases. Is the energy per flash changed? I suggest rewriting to first address the scaling for the flash rates, including the comment that the scaling factor is very similar to Finney et al (2014). Then discuss the scaling applied to get 5 Tg N per year globally.

L. 164 I think it would be good to include in the text what is said in the caption of Figure 1 regarding the satellite data are regridded to the model grid.

L. 174 The model ozone column is regridded. I assume that it is placed on the same grid as the satellite climatology (which is what in degrees latitude and longitude?). Could the sentence be clarified? "... is regridded to the satellite grid of x by y degrees and then compared on this grid. The model ozone column was not sampled the satellite track. (perhaps this last sentence is placed before the previous sentence).

L. 178 Hard to believe Thompson (2003) included data until 2011! It looks like 2011 should be 2000.

L. 179 Perhaps add values of latitudes for the 4 regions.

L. 187 What does "... extension of the evaluation over a smaller region ..." mean? I assume that this paper evaluates lightning over a larger region than what was used by

Finney et al. (2014).

L. 275 Insert "NOx" before emissions.

L. 303 Add "in the ZERO simulation" in stating which case has reduced deposition.

L. 305 is not clear. Is not the ZERO simulation corresponding to a reduction of N emissions by definition? That is, it is how the simulation is configured. What is the point of "less than the range of estimates for lightning emissions"?

L. 315 Use "whole" instead of "total" to be consistent with table.

L. 315-319 Why not just say "less than by 13 Tg" instead of "difference of -13 Tg"? I think your meaning may become clearer. Likewise, for the other differences stated in this paragraph.

L. 309-324. Consider revising the construction of this paragraph, which is making the point that location of the emissions (tropics versus extratropics) matters because production of O3 in the tropical upper troposphere will result in more O3 transported into the stratosphere. Previous studies found this result, and your results do as well. Implement basic paragraph construction: Topic of paragraph (or point being made), support of this topic, concluding sentence.

L. 326-333 Remind the reader that although the ICEFLUX and CTH simulations were designed to have the same magnitude of lightning flashes and lightning-NOx production, the location of the lightning and lightning-NOx differs between simulations, citing Figure 1 or other supporting information.

L. 355 add "by peroxy radicals".

L. 358-359. Change to "Ox precursors are transported downwind of convection before they form ozone".

L. 361-363. The last sentence of the paragraph should be the first sentence of the next paragraph.

L. 473. When the authors say, "at the time of emission", do they mean within the model time step? In other words, 15% of the Ox production associated with lightning occurs within 20 minutes of the lightning flash (or NO emission)?

Table 1. Add units for RMSE and mean bias.

Table 2. Add information about values in parentheses.

---

## Author Comment (AC1) · 23 May 2016

Response to reviewer 1

We thank the reviewer for their comments which have helped to improve and clarify several points in the manuscript. We are pleased that the reviewer found the manuscript to be well written and organised, and we address their comments below. We have included line numbers with respect to the revised marked-up document to accommodate matching changes in the manuscript in response to these comments.

Significant Comments:
Lines 98-99: Are there separate variables for cloud ice and precipitation-size ice (snow and graupel)? If so, please be more specific here.

In the UKCA climate-chemistry model the different aspects of cloud ice, including snow, pristine ice and riming particles, are all considered together as part of one prognostic cloud ice variable. We have added the sentence "The cloud ice variable includes snow, pristine ice and riming particles." (Line 108). Microphysical processes such as melting snow and riming are represented but the prognostic variable provides the bulk response of all these components as cloud ice. The references included in the text in this paragraph provide further details of the microphysical scheme.

Line 171: Are there any biases in this tropospheric ozone product relative to sondes or other satellite observations?

The tropospheric ozone product used in this paper is that of Ziemke et al. (2011). Ziemke et al. (2011) evaluated the MLS/OMI product against ozonesondes and an alternative satellite product which combined SAGE and MLS with ozonesondes to estimate tropospheric ozone. There is no apparent systematic under- or over- estimation across the sonde sites. We have added the following sentences discussing this evaluation: "In an evaluation against ozone sondes with broad coverage across the globe, the MLS/OMI product generally simulated the annual cycle well (Ziemke et al., 2011). The annual mean tropospheric column ozone mixing ratio of the MLS/OMI product was found to have a root mean square error (RMSE) of 5.0 ppbv, and a correlation of 0.83, compared to all sonde measurements. The RMSE was lower and correlation higher (3.18ppbv and 0.94) for sonde locations within the latitude range 25°S to 50°N." (Lines 207-212)

Lines 219-220: The correlation is also not improved with ICEFLUX in the southern extratropics.

The reviewer is correct. This paragraph has been revised to: "Overall, the ICEFLUX approach reduces the errors in the annual cycles of lightning. This scheme improves the correlation between simulated and observed lightning compared to CTH scheme in the northern extratropics and southern tropics. It has a lower correlation in the northern tropics, where both approaches for simulating lightning have difficulties, and in the southern extratropics, where the magnitude of the bias is much reduced upon compared to the CTH approach." (Lines 263-268).

Lines 236-237: It would benefit the paper if these statistics were presented for the latitude bands.

We present an extended version of Table 1 with latitude bands included below. The top section of this table is the original version which remains in the manuscript. The other sections show statistics for other latitudes bands which we have decided not to add to the manuscript. The adjusted RMSE for each region was calculated using the mean bias over the full 60S-60N region. While we agree that it was worth examining these statistics, we feel they do not significantly add to the manuscript. An additional conclusion could be that the

adjusted RMSE of the ZERO simulation is lower than that for the CTH and ICEFLUX simulations in northern midlatitudes. However, given the still comparatively large unadjusted RMSE in the region, this does not add sufficient value to merit inclusion of such a large amount of extra data.

| Latitude band | Run | r | RMSE (DU) | Mean bias (DU) | adjusted RMSE (DU) |
|---|---|---|---|---|---|
| Full 60S-60N | CTH | 0.82 | 5.5 | -2.8 | 4.1 |
|  | ICEFLUX | 0.84 | 5.7 | -3.2 | 3.9 |
|  | ZERO | 0.83 | 10.7 | -7.4 | 4.6 |
| 60N-30N | CTH | 0.65 | 4.0 | – | 3.6 |
|  | ICEFLUX | 0.67 | 3.8 | – | 3.5 |
|  | ZERO | 0.69 | 6.4 | – | 3.0 |
| 30N-Eq | CTH | 0.95 | 3.3 | – | 3.4 |
|  | ICEFLUX | 0.96 | 3.4 | – | 2.6 |
|  | ZERO | 0.96 | 9.2 | – | 2.7 |
| Eq-30S | CTH | 0.90 | 4.0 | – | 2.4 |
|  | ICEFLUX | 0.93 | 4.9 | – | 2.2 |
|  | ZERO | 0.93 | 12.3 | – | 5.1 |
| 30S-60S | CTH | 0.81 | 8.6 | – | 6.1 |
|  | ICEFLUX | 0.81 | 8.9 | – | 5.8 |
|  | ZERO | 0.81 | 13.5 | – | 6.3 |

We realise that we omitted an explicit description of the range of the MLS/OMI product in the manuscript (60S-60N), so we have added a comment to the data description section (Line 198) and the Table 2 caption.

Line 300: It is not clear how you are defining the tropopause. How are the 380K surface and the 2 PVU surface combined?

We have added a reference and modifed the text to read: "These budget terms are for the troposphere. Here, the tropopause is defined at each model time step using a combined isentropic-dynamical approach based on temperature lapse rate and potential vorticity (Hoerling1993)." (Line 391-395). The reference provides the details and motivation for the tropopause definition used in the UKCA model. It uses a thermal definition equatorward of 13 degrees, and a dynamical definition poleward of 28 degrees. In between there is a smooth transition using weightings of the two definitions. This definition of the tropopause overcomes issues with the individual definitions and produces a tropopause surface which is a continuous function of latitude.

Line 436: I am not sure what is the significance of this Ox production in the first 20 minutes. Isn't it primarily just the production of NO2 as it comes into equilibrium with NO in the

atmosphere after flashes occur? Very little ozone production is going to be produced in 20 minutes.

Following NO emission, the NO is oxidised to bring NO into equilibrium with NO2. This happens principally though the reaction of NO with O3. The Ox production term diagnosed here does not include this reaction flux since the NO2 product is also an Ox species – no Ox is produced or lost from this reaction. Instead the Ox production term is dominated by the oxidation of NO by peroxy radicals. A small proportion of this Ox production in the first 20 minutes will be associated with equilibration, but the remainder reflects equilibrium NOx cycling and consequent O3 production. We have added the following text: "oxidation of NO to NO2 by peroxy radicals." to the Ox budget description in Section 4 (Line 388). The Ox budget diagram (Figure 6) demonstrates, terms through the use of a grey arrows, that reactions converting O3 to NO2 and vice versa are not counted in the Ox budget.

Table 2: What are the percentages? Are they the percentage changes from the CTH scheme? More meaningful might be to show the percentage changes of CTH and ICEFLUX from ZERO.

Good point. The percentage differences shown in Table 2 are with respect to the CTH scheme. We have now added a statement to this effect to the table caption. We choose to show changes with respect to CTH to demonstrate both the small effect of horizontal changes in distribution (the ICEFLUX column) and the large effect of emission changes (the ZERO column). We agree that percentages with respect to ZERO are useful but feel that percentages with respect to CTH make the above point most clearly.

Figure 5: There should be stratospheric influx of NO2 and other NOy species.

We thank the reviewer for highlighting this. The figure referred to is figure 6 in the new manuscript. The stratospheric influx is an inferred quantity which is derived to balance the other budget terms. It is not calculated for each species individually but is a value for the influx of total Ox, though this is dominated by the O3 contribution. We felt that the most accurate way to portray this was by including a stratospheric influx arrow pointing to the Ox label, instead of any individual species. We have added a statement describing the stratospheric influx term in the figure 6 caption.

Minor Comments:
Line 21: comparison of models

Changed to "between". Line 22

Line 183: define ACCMIP

Changed. Line 224

Line 267: .....instead the correlation values between the model and the sonde data (Figure 3) provide a more......

Changed. Line 356

Line 302: ....production and losses when lightning is added (Table 2).

Changed to "when lightning NOx emissions are removed". Line 397

Line 446: The increase is linear up to approximately 0.006 fl km-2 min-1 and then becomes steeper up to o,02 fl km-2 min-1 at which....

We show below Figure 11A plotted with a linear x-axis which highlights the linear features which change at approximately 0.02 fl. km$^{-2}$ 20min$^{-1}$. We have revised the text to read: "A linear increase in Ox production is apparent up to approximately 0.02 fl. km$^{-2}$ 20min$^{-1}$ at which point the two schemes produce 1 to 1.5 kg km$^{-2}$ 20min$^{-1}$ of Ox. Beyond this point, the Ox production simulated by the ICEFLUX approach increases still linearly but with a shallower gradient." (Lines 568 - 570)

---

## Author Comment (AC2) · 23 May 2016

We thank reviewer 2 for their comments which clarified the manuscript and included useful suggestions on how to expand the analysis to be more relevant to future aircraft studies. We are very pleased that they suggest our new lightning parametrisation analysis would be an important addition to the literature. We address their comments below. We have included line numbers with respect to the revised marked-up document to accommodate matching changes in the manuscript in response to these comments.

Specific Comments

Title. To convey what kind of parameterization, I would suggest modifying the title to say, "using a new lightning parameterization".

The title has been changed to "…using a new global lightning parametrisation"

L. 126. How does the lightning-NOx scheme differentiate between cloud-to-ground (CG) and intracloud (IC) lightning? Does it need to make this difference if the production of NO per flash is the same for CG and IC flashes and the vertical distribution of NO sources is the same for CG and IC flashes?

Whilst the UKCA model does apply a method to determine the IC:CG ratio, it has no consequence in this study because, as the reviewer points out, we choose to have equal NO production for both types and because the vertical distribution is based on that of Ott et al. (2010) which is not dependent on the IC:CG ratio. A sentence has been added to clarify this: "As a consequence, the distinction between cloud-to-ground and cloud-to-cloud has no effect on the distribution or magnitude of lightning NOx emissions in this study." (Line 149).

Section 3. Both lightning flash rate schemes depend on how well the model predicts cloud top height or ice mass flux. Has there been an attempt to evaluate these parameters (or a proxy) from the chemistry-climate model to a climatology of reanalysis data?

Several evaluations of the representation of clouds in the Unified Model have been performed in the literature and references for these have been added to Section 2.1, along with a paragraph describing the relevant results (Lines 114-127). No specific evaluation was carried out for this study since the upward cloud ice flux cannot be well constrained by observations on a global scale. We can infer that, given the similar distribution of lightning flash rates to the study of Finney et al. (2014), as well as similar total annual flashes, the upward cloud ice flux at 440hPa simulated in the Unified Model used here is similar to the ERA-Interim reanalysis data used by Finney et al. (2014).

Section 3.1 has a nice analysis of lightning flash rates in different latitude bands, and remarks upon differences in continental regions. I wonder if an additional figure showing how the model performs for different continents could be included and discussed. For example, showing the annual cycle of North American, South American, African, India and East Asia, and Australia (and maybe tropical oceanic region) lightning should give peaks at different times of the year. This type of figure would be a natural follow on to Figure 1 because the eye is drawn to each of these regions when viewing Figure 1.

We agree that focus on specific regions would be useful and we have therefore included six such regions in a new Figure 3 in the revised manuscript. We have also added text to discuss the new figure (Lines 269-307), as well as revised the previous paragraph over lines 249-362 to improve the flow of the text. Below, we show a map of land regions and annual cycles of regional-mean lightning flash rates for potential regions that were initially considered. We made a selection from these in order to offer an interesting but succinct discussion on regional performance.

[Figure]

L. 252. While both lightning-NOx schemes show a general underestimate of ozone in the middle and upper troposphere of the tropics, they are both within the variability of the observations (while no lightning-NOx is outside that variability). In fact, the northern tropics appears to have quite good agreement. If you want to point out the underestimation, restrict the comment to the southern tropics. Second, what is the variability in the model results?

We agree with the reviewer, although we also feel it worth noting in the text that the spring peak in the northern tropics is underestimated. The reviewer's point regarding the variability range is also correct. We have modfied the text to read "Both schemes still show an underestimate compared to observations all year round in the southern tropics and during

spring in the northern tropics, but are within the variability of sonde measurements." (Line 339-341). There is no estimate of variability for the models as they are based on a single year run, however, by using a climatology of SSTs and emissions from year 2000, the simulation should broadly represent a tropospheric ozone climatology.

L. 284. I like the conclusion from the analysis that point to April and October as specific months to focus field campaigns. However, aircraft field campaigns can only cover a region (and not a latitude band). Can you recommend where field campaigns should focus? A similar analysis of continental regions would be helpful.

We feel that the additional analysis of the lightning annual cycle of particular regions (Figure 3) provides useful insights regarding this point. The biases in lightning in the southern tropics have been identified as originating in biases in northern South America, so this would be an appropriate region for studies. We have added a sentence to the ozone sonde discussion which refers back to this: "It may be of particular use for field campaigns studying the chemical impact of lightning to focus on these months and, as discussed in Section 3.1, South America could provide a useful region in which to develop understanding of lightning activity and therefore also its impacts on tropospheric ozone." (Line 375-378).

L. 297. I assume that the major Ox production is through oxidation of NO by peroxy radicals. This should be clarified to avoid confusion with NO + O3 producing NO2. It is curious that Table 2 discusses production and loss rates of Ox, but burdens of O3. I assume that is because O3 is the dominant Ox species (although it is of equal size to NO2 and O(1D) in Figure 5). It would be good to clarify in this paragraph why you discuss O3 burdens juxtaposed with the Ox production and loss rates discussion.

We thank the reviewer for this comment. We have added "by peroxy radicals" and a sentence to explain that only the ozone burden is considered because it makes up the majority of the Ox burden (Lines 388-390). The NO2 and O$^1$D species had been highlighted in the Ox budget diagram because they were involved in the production and loss key fluxes. However, to maintain a consistent approach to that used with the highlighting of reactions in Figure 6, the NO2 and O1D species have now been shaded grey along with the other Ox species apart from O3.

L. 303-304. Perhaps the characterization of the ZERO case could be revised. I think it should be described as the following. There is less production of Ox (or O3) without lightning-NOx emissions, resulting in a smaller O3 burden and therefore reduced Ox losses and shorter O3 lifetime. Can anything be said about linear or non-linear responses? For example, it seems that the lifetime decrease is less than the Ox loss rate decrease, and both are less than the decrease in Ox production.

This is a good point. The text has been modified to make the description of budget changes clearer (Line 398-403). We have commented that the lifetime changes by less and the reason for this, as we agree that this is an important point. However, to determine linearity, multiple experiments with different emissions would be needed.

Section 4. In addition to the comparison of the ZERO case with the two other cases, there should be a statement pointing out similarities between ICEFLUX and CTH, including the point at the beginning of Section 5.

An additional sentence has been added: "The largest differences between the Ox budgets of the ICEFLUX and CTH approaches are in the ozone burden and lifetime but these are only 2%." (Line 413). At the beginning of section 5, an additional sentence has been added to point out that the two schemes produce similar values for the global Ox budget (Lines 438-440).

L. 341-344. I think it would be helpful to the reader to repeat how the NO lightning emissions are placed vertically for each scheme. It is also not clear to me how the horizontal distribution affects the vertical distribution. My interpretation is that ICEFLUX predicts lower lightning-NOx emissions in the tropics based on the storm parameters and more in the extratropics. While the magnitude of NO emissions is less in the ICEFLUX scheme for the tropics, those emissions are still distributed according to the Ott et al. (2010) curves to cloud top height (lines 126-128; and cloud top height should be the same in the two simulations). However, I think the authors are trying to say that the ICEFLUX scheme produces a lot of lightning-NO emissions in storms with lower cloud tops. There is also the point that because the CTH scheme has greater NO emissions in taller clouds, there is a substantial difference in where the NO emissions are found vertically. I think this could easily be supported by a plot of lightning-NO emissions versus cloud top height for different latitude bands.

The method to distribute LNOx vertically has been restated (Lines 453-455) and we have included a reference to the model parametrisation description in section 2.2. Regarding how the horizontal distribution of lightning affects the vertical distribution of LNOx emissions, we agree with the reviewer's interpretation and discuss the point below. The new section in the manuscript reads: "As described in section 2.2, the column LNOx is distributed up to the cloud-top, and this is how a coupling exists between the horizontal LNOx distribution simulated by the CTH approach and the height that LNOx emissions reach. This means that, by basing the horizontal lightning distribution on cloud-top height and then distributing emissions to cloud top, LNOx is most effectively distributed to higher altitudes." (Lines 453-457).

The vertical LNOx distribution is determined by the cloud top height at those (horizontal) locations where lightning has been diagnosed to occur. If cloud-top height is used to determine the horizontal distribution of lightning then the highest emissions will occur where the cloud tops are highest and therefore those emissions will be distributed to as high a level as possible for the given set of modelled cloud top heights.

An illustration of this is shown below for two storms (ultimately grid cells in the model): one with the highest cloud top, and one with the highest upward ice flux. Assuming the largest emission for the CTH approach is comparable to the largest emission for the ICEFLUX approach, the emissions are skewed to lower altitudes where using the ICEFLUX parametrisation. Figure 7E shows the gross Ox production zonal-altitudinal distribution and demonstrates that in the northern tropics there is a shift of Ox production to lower altitudes reflecting emissions at lower altitudes.

[Figure]

Regarding the reviewer's final suggestion, a plot of lightning NO emissions against cloud-top height would show the relationship used by the CTH approach scaled by the NO per flash parameter. For the ICEFLUX approach, it would show a different relationship because cloud top height is not used to determine the size of emission. This second plot would approximately show the relationship between upward ice flux and cloud-top height in the model. We feel that the inclusion of such plots would be somewhat of a digression and would not make the point clearer to the reader.

L. 369-375. Could this be clarified? It was already established that a reduction in Ox production decreased O3 mixing ratios and therefore Ox loss rates (Section 4). However, in these lines it says there is an increase in Ox production in the middle and lower troposphere but a reduction of O3 concentrations, when comparing ICEFLUX and CTH results. Is Ox partitioned differently, meaning there is more HNO3 that can be removed? What loss process dominates (O3 chemical loss or Ox wet deposition)?

The reasoning in the paragraph (Lines 485-492) is that there is an increase in net chemical production but that this is because the chemical loss is reduced in the region due to reduced ozone concentrations. Ozone concentrations have reduced because less ozone is produced in the upper troposphere to be transported within the tropics in the Hadley cell. The paragraph has been modified to focus on the upper tropospheric change impacting the whole tropical troposphere (Lines 485-492). The NOy wet deposition is not discussed but given there is no change in the global total deposition and NOy wet deposition makes up a small proportion of total deposition terms, it is reasonable to assume that it isn't a key driver in the ozone concentration distribution.

L. 405-410. I was surprised that the ICEFLUX lightning flash rate frequency distribution was not discussed. Also, although it is not the point of section 5, I wonder if it would be useful to include LIS/OTD frequency distribution in Figure 9.

The LIS/OTD product used here is a monthly climatology, hence cannot be compared to the 20 minute flash rates simulated by the model shown in in Figure 9. Whilst the individual LIS/OTD observations could be used to produce a frequency distribution, because the satellites do not measure all locations simultaneously, much care would be required to

sample the model at the same locations and times. We therefore choose to describe the CTH distribution relative to the ICEFLUX distribution. An additional sentence has been added to give some context to the text on the ICEFLUX distribution: "The ICEFLUX approach produces a similar distribution to that produced by the same scheme applied in the study by Finney et al. (2014.) In that study the ICEFLUX frequency distribution had a fairly average distribution compared to four other lightning parametrisations with slightly more occurrences of low flash rates." (Lines 527-530).

L. 460. It is an interesting finding that Ox production efficiency is less for higher flash rates (at least initial Ox production). Could the authors speculate why this would happen? Or suggest analysis that could be done in order to explain why. I would imagine the HO2 and RO2 abundance might play a role. Are there connections between flash rate and location to VOC sources? For example, Barth et al. (2012) showed more O3 produced from storms occurring over VOC-rich regions (e.g. southeast U.S.).

This is definitely an interesting point. When extra NO is added there is a corresponding increase in Ox production. However, each additional NO molecule produces less Ox, as NOx cycling becomes less efficient for higher NOx levels. This is to be expected because other species involved in the NOx cycle such as HO2/VOCs do not increase in step with Ox to maintain the same production efficiency. Whilst location and background levels of ozone precursors play a role, the figures are based on global data and shows that this relationship holds for LNOx in general, though obviously high LNOx will have a tendency to occur in particular hotspots. The Barth et al. (2012) paper is a useful example of how VOCs interact with LNOx to affect ozone production. A reference to Barth et al. (2012) and three extra sentences have been added: "This suggests that as the NO increases, NOx cycling and therefore ozone production decreases in efficiency. This is likely a result of peroxy radical availability and VOC abundance limiting the rate of NOx cycling. Evidence for such control of VOC precursors on ozone production in US thunderstorms has been presented by Barth et al. (2012)." (Lines 585-588).

L. 465. How did the authors translate the Ox production efficiencies to Ox produced per mole of NO?

We have added the following at the beginning of the sentence "Using the NO production per flash of 250 mol(NO) fl.$^{-1}$ stated in Section 2.2,…" (Line 591)

L. 477. Here, the authors argue that more Ox is produced by the CTH scheme because NOx has a longer lifetime at higher altitudes. However, the analysis is for the initial Ox production ("at the time of emission")? How does the NOx lifetime affect the Ox production shown in Figure 10, which is "at the time of emission"?

This is a good point. There are several factors that can lead to increased ozone production efficiency from NOx at higher altitudes: the longer lifetime of NOx, the rate of NOx cycling, and efficiency of NOx cycling. The focus of the original text on the lifetime of NOx in this case was too specific as the lifetime will not play a substantive role in determining the ozone production efficiency over the initial 20 minute time step. We have amended the text to say "where ozone production efficiency is greater", since the reason for this is a combination of these factors (Line 606). The text on "NOx lifetime" mentioned earlier in the section and has also been broadened (Line 575).

Technical Comments
L. 9 Insert "NO" before emission.

Changed. Line 8

L. 17 Replace "-" with ";"

Changed. Line 17

L. 16-18 I suggest adding a caveat that more ozone production can subsequently occur from the high flash rate regions.

The term "initially" Ox production is added to the abstract (Lines 18). In the Section 6, we have added additional text "This study has analysed the Ox production occurring in the first 20 minutes, but further Ox production can occur over longer time periods." (Lines 616)

L. 21-22 Change to "for comparison between models and observations : : :".

Changed. Line 22

L. 27 NO2 lifetime may be shorter in the upper troposphere because its photolysis rate is greater. I think it would be better to rewrite the sentence to say NOx lifetime is longer in the upper troposphere (rather than the individual species).

Changed. Lines 28

L. 51 Could a reference be cited supporting that the upper troposphere is the region with most efficient ozone production?

We add the reference of Dahlmann et al. (2011) (Line 53) which addresses the ozone production efficiency of different sources including lightning and aircraft NOx and finds that these two sources have a greater ozone production efficiency because of their location.

L. 53 Please delete "simplified". I find cloud chemistry models to be rather complex.

This has now been removed. (Line 55)

L. 63-64 It would be better written as, ": : : of low flash rates, which are unrealistic compared to observed flash rates. This results in low NOx concentrations and greater ozone production efficiency : : :."

Changed. Lines 64-66

L. 86 Please add more information about the chemistry represented in the model. Is it the "standard troposphere" chemistry or does it have the added isoprene chemistry, both described in O'Conner et al. (2014)? I suggest including number of species, stating it describes methane, ethane, and propane (and maybe isoprene) hydrocarbon chemistry.

More information has been added which addresses the comment. Isoprene chemistry is included and appropriate references are given. Lines 90-94

L. 147-151 Could this be rewritten? It appears that only lightning flash rates are scaled to obtain a global values of 46 fl/s, because the NO production per energy is the same for both cases. Is the energy per flash changed? I suggest rewriting to first address the scaling for the flash rates, including the comment that the scaling factor is very similar to Finney et al (2014). Then discuss the scaling applied to get 5 Tg N per year globally.

A scaling factor is calculated for each parametrisation to achieve the same global annual flash rate. Each flash has equal energy. Then the NO production per Joule is chosen in order to produce 5 TgN per year given the total number of flashes (which is the same for

each parametrisation). The ordering and wording of the sentences has been altered to make this clearer: "However, for this study we choose to have the same flash rate and global annual NOx emissions for both schemes. A scaling factor was used for each parametrisation that results in the satellite estimated flash rate of 46 fl./s, as given by Cecil et al. (2012). … Given that each parametrisation produces the same number of flashes each year and each flash has the same energy, a single value for NO production can be used. As above, a value of 12.6 X $10^{16}$ NO molecules $J^{-1}$ was used for both schemes which results in a total annual emission of 5 TgN $yr^{-1}$." (Lines 171-182)

L. 164 I think it would be good to include in the text what is said in the caption of Figure 1 regarding the satellite data are regridded to the model grid.

Changed. Lines 194-195

L. 174 The model ozone column is regridded. I assume that it is placed on the same grid as the satellite climatology (which is what in degrees latitude and longitude?). Could the sentence be clarified? ": : : is regridded to the satellite grid of x by y degrees and then compared on this grid. The model ozone column was not sampled the satellite track. (perhaps this last sentence is placed before the previous sentence).

The model is regridded to the MLS/OMI grid of 5x5 degrees. The sentence has been rephrased as "In Section 3.2, the simulated annual mean ozone column is regridded to the MLS/OMI grid of 5° by 5° and compared directly to the satellite climatology without sampling along the satellite track." (Lines 204-206)

L. 178 Hard to believe Thompson (2003) included data until 2011! It looks like 2011 should be 2000.

The Thompson [et al. *now corrected*] (2003) paper describes the sites but this data set has since been extended. The sentence has been revised to say this. Lines 214-216

L. 179 Perhaps add values of latitudes for the 4 regions.

Added. Lines 218

L. 187 What does ": : : extension of the evaluation over a smaller region : : :" mean? I assume that this paper evaluates lightning over a larger region than what was used by Finney et al. (2014).

Yes, the region used in Finney (2014) was smaller. This sentence has been revised for clarity. Line 228

L. 275 Insert "NOx" before emissions.

Changed. Line 364

L. 303 Add "in the ZERO simulation" in stating which case has reduced deposition.

Referred to ZERO at the beginning of the sentence. Line 398

L. 305 is not clear. Is not the ZERO simulation corresponding to a reduction of N emissions by definition? That is, it is how the simulation is configured. What is the point of "less than the range of estimates for lightning emissions"?

The difference in LNOx (5 TgN/yr) is similar in magnitude to the uncertainty in the total lightning NOx source (~6 TgN/yr based on 2-8 TgN/yr). The sentences have been modified to try and make the point clearer: "There is uncertainty in the global lightning NOx source of 2-8 TgN emissions (Schumann and Huntrieser 2007), and there will be an associated uncertainty in the Ox budgets. Using no lightning (ZERO) corresponds to a reduction of 5 TgN emissions over the year - less than the range of uncertainty in LNOx. " Lines 404-407

L. 315 Use "whole" instead of "total" to be consistent with table.

Changed for all instances in the paragraph. Lines 415,420,422 and 426

L. 315-319 Why not just say "less than by 13 Tg" instead of "difference of -13 Tg"? I think your meaning may become clearer. Likewise, for the other differences stated in this paragraph.

Changed. Lines 423-428

L. 309-324. Consider revising the construction of this paragraph, which is making the point that location of the emissions (tropics versus extratropics) matters because production of O3 in the tropical upper troposphere will result in more O3 transported into the stratosphere. Previous studies found this result, and your results do as well. Implement basic paragraph construction: Topic of paragraph (or point being made), support of this topic, concluding sentence.

Agreed. The beginning of the paragraph has been altered. Line 415

L. 326-333 Remind the reader that although the ICEFLUX and CTH simulations were designed to have the same magnitude of lightning flashes and lightning-NOx production, the location of the lightning and lightning-NOx differs between simulations, citing Figure 1 or other supporting information.

The following sentence has been added, "In the previous section, it was demonstrated that the global tropospheric Ox budget is affected principally by the magnitude of emissions and not the location of emissions. This was achieved by using the same total emissions but different distributions of lightning in the CTH and ICEFLUX approaches (Figure 1), which simulate little difference in the global Ox budget terms." Lines 436-440

L. 355 add "by peroxy radicals".

Changed. Line 469

L. 358-359. Change to "Ox precursors are transported downwind of convection before they form ozone".

Amended the sentence to: "Furthermore, ozone precursors are transported downwind of convection before they form ozone.". Line 473

L. 361-363. The last sentence of the paragraph should be the first sentence of the next paragraph.

Changed. Line 477

L. 473. When the authors say, "at the time of emission", do they mean within the model time step? In other words, 15% of the Ox production associated with lightning occurs within 20 minutes of the lightning flash (or NO emission)?

Yes, in the model time step including the emission. This has been clarified in this instance and at the first use of *initial*. Lines 557 and 601

Table 1. Add units for RMSE and mean bias.

Added.

Table 2. Add information about values in parentheses.

Added to Table caption.